

# Comparison of the GOSAT TANSO-FTS TIR $CH_4$ volume mixing ratio vertical profiles with those measured by ACE-FTS, ESA MIPAS, IMK-IAA MIPAS, and 16 NDACC stations

Kevin S. Olsen[1], Kimberly Strong[1], Kaley A. Walker[1,2], Chris D. Boone[2], Piera Raspollini[3], Johannes Plieninger[4], Whitney Bader[1,5], Stephanie Conway[1], Michel Grutter[6], James W. Hannigan[7], Frank Hase[4], Nicholas Jones[8], Martine de Mazière[9], Justus Notholt[10], Matthias Schneider[4], Dan Smale[11], Ralf Sussmann[4], and Naoko Saitoh[12]

[1]Department of Physics, University of Toronto, Toronto, Ontario, Canada
[2]Department of Chemistry, University of Waterloo, Waterloo, Ontario, Canada
[3]Istituto di Fisica Applicata "N. Carrara" (IFAC) del Consiglio Nazionale delle Ricerche (CNR), Florence, Italy
[4]Institut für Meteorologie und Klimaforschung, Karlsruhe Institute of Technology, Karlsruhe, Germany
[5]Institute of Astrophysics and Geophysics, University of Liège, Liège, Belgium
[6]Centro de Ciencias de la Atmósfera, Universidad Nacional Autónoma de México, Mexico City, Mexico
[7]Atmospheric Chemistry Division, National Center for Atmospheric Research, Boulder, CO, USA
[8]Centre for Atmospheric Chemistry, University of Wollongong, Wollongong, Australia
[9]Belgisch Instituut voor Ruimte-Aëronomie–Institut d'Aéronomie Spatiale de Belgique (IASB-BIRA), Brussels, Belgium
[10]Institute for Environmental Physics, University of Bremen, Bremen, Germany
[11]National Institute of Water and Atmospheric Research Ltd (NIWA), Lauder, New Zealand
[12]Center for Environmental Remote Sensing, Chiba University, Chiba, Japan

*Correspondence to:* K. S. Olsen (ksolsen@atmosp.physics.utoronto.ca)

**Abstract.** The primary instrument on the Greenhouse gases Observing SATellite (GOSAT) is the Thermal And Near infrared Sensor for carbon Observations (TANSO) Fourier Transform Spectrometer (FTS). TANSO-FTS uses three short-wave infrared (SWIR) bands to retrieve total columns of $CO_2$ and $CH_4$ along its optical line-of-sight, and one thermal infrared (TIR) channel to retrieve vertical profiles of $CO_2$ and $CH_4$ volume mixing ratios (VMRs) in the troposphere. We examine version 1 of

5   the TANSO-FTS TIR $CH_4$ product by comparing co-located $CH_4$ VMR vertical profiles from two other remote sensing FTS systems: the Canadian Space Agency's Atmospheric Chemistry Experiment-FTS (ACE-FTS) on SCISAT (version 3.5), and the European Space Agency's Michelson Interferometer for Passive Atmospheric Sounding (MIPAS) on Envisat (ESA ML2PP version 6 and IMK-IAA reduced-resolution version V5R_CH4_224/225), as well as 16 ground stations with the Network for the Detection of Atmospheric Composition Change (NDACC). This work follows an initial inter-comparison

10  study over the Arctic, which incorporated a ground-based FTS at the Polar Environment Atmospheric Research Laboratory (PEARL) at Eureka, Canada, and focuses on tropospheric and lower-stratospheric measurements made at middle and tropical latitudes between 2009 to 2013 (mid 2012 for MIPAS). For comparison, vertical profiles from all instruments are interpolated onto a common pressure grid, and the ACE-FTS, MIPAS, and NDACC vertical profiles are smoothed using the TANSO-FTS averaging kernels. We present zonally-averaged mean $CH_4$ differences between each instrument and TANSO-FTS with

15  and without smoothing, examine their information content, sensitive altitude range, correlation, a priori dependence, and the



variability within each data set. Partial columns are calculated from the VMR vertical profiles, and their correlations are examined. We find that the TANSO-FTS vertical profiles agree with the ACE-FTS and both MIPAS retrievals' vertical profiles within 4 % below 15 km when smoothing is applied to the profiles from instruments with finer vertical resolution, but that the relative differences can increase to on the order of 25 % when no smoothing is applied. Computed partial columns are tightly correlated for each pair of data sets. We investigated whether the difference between TANSO-FTS and other $CH_4$ VMR data products varies with latitude. Our study reveals a small dependence of around 0.1 % per ten degrees latitude, with smaller differences over the equator, and greater differences towards the poles.

# 1 Introduction

The Greenhouse gases Observing SATellite (GOSAT) was developed by Japan's Ministry of the Environment (MOE), National Institute for Environmental Studies (NIES), and the Japan Aerospace Exploration Agency (JAXA), and was launched in 2009 with an inclination of $98°$ (Yokota et al., 2009). The objectives of the GOSAT mission include monitoring the global distribution of greenhouse gases, estimating $CO_2$ source and sink locations and strengths, and verifying the reduction of greenhouse gas emissions, as mandated by the Kyoto Protocol. GOSAT carries two instruments: the Thermal And Near infrared Sensor for carbon Observations (TANSO) Fourier Transform Spectrometer (FTS) and the TANSO Cloud and Aerosol Imager (TANSO-CAI).

TANSO-CAI is a radiometer with four spectral bands between 0.37 and 1.6 μm, each around 0.02 μm wide and chosen to avoid $H_2O$ and $O_2$ absorption. TANSO-CAI is able to measure the cloud fraction in the field-of-view of TANSO-FTS (Ishida and Nakajima, 2009; Ishida et al., 2011). TANSO-FTS is a nadir-viewing double-pendulum FTS, whose technical details are described in Sect. 2.1. TANSO-FTS makes observations of infrared radiation emitted from the Earth's atmosphere in four bands. Three bands are in the short-wave infrared region and are used to measure total columns of $CO_2$ and $CH_4$. The fourth channel is in the thermal infrared (TIR) to provide GOSAT with sensitivity to the vertical structure of $CO_2$ and $CH_4$.

This work follows Holl et al. (2016), who compared Atmospheric Chemistry Experiment (ACE) FTS version 3.5 (v3.5) and TANSO-FTS TIR version 1 (v1) vertical profiles with those measured by a ground-based FTS at the Polar Environment Atmospheric Research Laboratory (PEARL) in Eureka, Canada. We employ a similar methodology, extend that study globally, and include multiple ground-based FTSs that are part of the Network for the Detection of Atmospheric Composition Change (NDACC). Holl et al. (2016) observed that after smoothing the ACE-FTS profiles using the TANSO-FTS averaging kernels and a priori profiles, the difference is close to zero above 15 km, but that there is a bias at lower altitudes where TANSO-FTS retrieves more $CH_4$, with a mean excess of 20 ppbv in the troposphere.

Our objective is to investigate whether the results of Holl et al. (2016) are local, or hold at all latitudes, and to provide additional global validation of the TANSO-FTS v1 $CH_4$ data product. Any biases in the v1 data product need to be well understood for it to be used by other researchers, and their discovery may lead to improvements of future versions.

In this manuscript, we examine the TIR data product from TANSO-FTS, specifically, $CH_4$ volume mixing ratio (VMR) vertical profiles, by determining when TANSO-FTS TIR retrievals of $CH_4$ were made in coincidence with those of other satellite-





borne and ground-based FTS instruments. Comparisons of satellite instruments are made with the ACE-FTS on SCISAT, described in Sect. 2.2, and the Michelson Interferometer for Passive Atmospheric Sounding (MIPAS) on the Environmental Satellite (Envisat), described in Sect. 2.3. The NDACC InfraRed Working Group (IRWG) has a network of ground-based FTSs; we used 16 that retrieve vertical profiles of $CH_4$ VMR to compare with the TANSO-FTS TIR data. The NDACC data

are described in Sect. 2.4. A summary of the instruments used in this study is given in Table 1.

For each comparison pair, the averaging kernels, information content, and variability of the retrievals are examined in Sects. 3 and 5. The instrument with finer vertical resolution is smoothed using the averaging kernels of the instrument with coarser vertical resolution (TANSO-FTS in all cases presented here). For each coincident pair, the absolute and relative differences of the smoothed and unsmoothed VMR vertical profiles are found and their means, correlation coefficients, $R^2$, and numbers

of coincident pairs are computed at each pressure level. For each vertical profile in a coincident pair, an overlapping vertical extent is selected using the sensitivity, or response, of the TANSO-FTS retrieval (area of the averaging kernel matrix), partial columns are computed over this range, and their correlations are examined. Finally, this altitude range is used to estimate the mean VMR difference taken over the vertical range for each coincident pair of profiles. This dataset shows any biases related to latitude, or any other parameters of the TANSO-FTS retrieval, such as incidence angle or surface type (land or water).

Sect. 4 describes the methods and criteria for determining coincident measurements between TANSO-FTS and each instrument. Sect. 6.1 provides a detailed description of the comparison methodology. Comparison results for each instrument are presented in Sect. 6.2. The satellite instruments are zonally averaged and each NDACC site is shown. Partial column calculation methodology is presented in Sect. 7.1 and correlation results are shown in Sect. 7.2. A discussion follows in Sect. 8, focusing on our investigation of biases within the TANSO-FTS retrievals related to latitude and other parameters.

## 2    Data sets

### 2.1    TANSO-FTS

TANSO-FTS makes measurements of radiance in four bands: 12900–13200 $cm^{-1}$, 5800–6400 $cm^{-1}$, 4800–5200 $cm^{-1}$, and 700–1800 $cm^{-1}$. The fourth band is in the TIR and is used to retrieve vertical profiles of $CH_4$ VMRs. TANSO-FTS has a spectral resolution of $0.2\,cm^{-1}$ and operates in a nadir or near-nadir viewing geometry (Kuze et al., 2009). To improve

coverage, its field of view sweeps longitudinally, and TANSO-FTS makes several measurements along each cross track, five measurements prior to August 2010, and three since then (Kuze et al., 2012). This leads to TANSO-FTS having the highest density of measurements and greatest coverage among the instruments considered herein.

Retrievals of v1 $CH_4$ follow the methodology for v1 $CO_2$ presented in Saitoh et al. (2009, 2016). They are performed on a fixed pressure grid and the pressure levels are adjusted based on the averaging kernels for the retrieval. In the v1 retrieval

algorithm, water vapour, nitrous oxide, ozone concentrations, temperature, surface temperature and surface emissivity were retrieved simultaneously with $CH_4$ concentration from V161.160 L1B spectra. A priori data are based on simulated data from the NIES transport model (Maksyutov et al., 2008; Saeki et al., 2013) and the retrievals use the HITRAN 2008 linelist (Rothman et al., 2009) with several updates up to 2011.





An initial comparison of TANSO-FTS v1 to a single NDACC station, Eureka, and to ACE-FTS measurements made in the Arctic within a quadrangle surrounding PEARL (60–90° N and 120–40° W) has been recently made (Holl et al., 2016). The v1

CH$_4$ product was also compared globally with the version 6 CH$_4$ data product from the Atmospheric Infrared Sounder (AIRS) on Aqua (Zou et al., 2016).

## 2.2 ACE-FTS

ACE-FTS was launched into a circular low-Earth orbit with an inclination of 74° in 2003 onboard the Canadian Space Agency's (CSA's) SCISAT. SCISAT also carries the ACE-Measurement of Aerosol Extinction in the Stratosphere and Troposphere Re-

trieved by Occultation (MAESTRO) instrument, a dual spectrophotometer with a wavelength range of 285–1030 nm and a spectral resolution of 1–2 nm. The scientific objectives of ACE are to study ozone distribution in the stratosphere, the relationship between atmospheric chemistry and climate change, the effects of biomass burning on the troposphere, and the effects of aerosols on the global energy budget (Bernath, 2017).

ACE-FTS is a high-resolution, double-pendulum FTS with a spectral resolution of 0.02 cm$^{-1}$ that covers a broad spectral

range between 750–4400 cm$^{-1}$. It operates in solar occultation mode, making a series of measurements for tangent altitudes down to 5 km (or cloud tops) at local sunrise and sunset along its orbital path (Bernath et al., 2005). Its level 2 data products are vertical profiles of temperature, pressure, and the VMRs of 36 trace gases, as well as isotopologues of major species, reported on an altitude grid at the measurement tangent altitudes or interpolated onto a 1 km grid. Retrievals of the Version 2.2 (v2.2) data product are described in Boone et al. (2005) and updates regarding the latest release, Version 3.5 (v3.5), are described in

Boone et al. (2013). V3.5 retrievals, with the data quality flags (v1.1) described in Sheese et al. (2015), are used herein.

When performing trace gas retrievals, tangent altitudes for each observation and vertical profiles of temperature and pressure are also retrieved using spectral fitting (not simultaneously). Comparisons with TANSO-FTS are made on a pressure grid using the retrieved pressure values at the ACE-FTS measurement heights. A priori temperature and pressure for ACE-FTS are derived from the NRL-MSISE-00 model (MSIS) (Picone et al., 2002), and from meteorological data provided by the

Canadian Meteorological Centre and their Global Environmental Multiscale (GEM) model (Côté et al., 1998). Fitted spectra are computed using the HITRAN 2004 spectral linelist (Rothman et al., 2005) with modifications described in Boone et al. (2013).

Validation of v2.2 CH$_4$ VMR vertical profiles is presented in de Mazière et al. (2008) and was performed using several ground-based FTSs that are part of NDACC, as well as one at Poker Flat. For that comparison, partial columns were com-

puted from the ACE-FTS CH$_4$ profiles and the correlation between partial columns computed from ground-based FTSs and from ACE-FTS was investigated. Validation was also done against the balloon-borne SPIRALE (Spectroscopie Infra-Rouge d'Absorption par Lasers Embarqués), the Halogen Occultation Experiment on the Upper Atmosphere Research Satellite, and MIPAS. de Mazière et al. (2008) determined that the ACE-FTS v2.2 data are accurate to within 10 % in the upper troposphere and lower stratosphere and to within 25 % at high altitudes. More recently, Jin et al. (2009) compared CH$_4$ from the Canadian Middle Atmosphere Model (CMAM) with measurements from ACE-FTS, the Sub-Millimeter Radiometer (SMR) on Odin and the Microwave Limb Sounder (MLS) on Aura, and found agreement with ACE-FTS within 30 %. Updates to the ACE-FTS



validation effort using v3.0 data and a description of the differences between v2.2 and v3.0 are presented in Waymark et al.
(2013).

## 2.3 MIPAS

MIPAS is a limb-sounding FTS that was placed in polar (inclination of $98°$) low-Earth orbit in 2002 onboard the European
Space Agency's (ESA's) Envisat. MIPAS aimed to provide global observations, during both night and day, of changes in the
spatial and temporal distributions of long- and short-lived species, temperature, cloud parameters and radiance. The instrument
was intended to have a maximum spectral resolution of $0.025\,\mathrm{cm}^{-1}$ (Fischer et al., 2008), but the slide system for the interfer-
ometer mirrors encountered a problem in 2004 and observations used in this study were made with a reduced effective spectral
resolution of $0.0625\,\mathrm{cm}^{-1}$, but finer vertical sampling. Further complications arose in 2012 and ESA lost communication with
Envisat, ending the mission.

The spectral range of MIPAS is $685–2410\,\mathrm{cm}^{-1}$, allowing the retrieval of multiple trace gases. MIPAS spectra are processed
independently by four research groups (Raspollini et al., 2014). In this paper, we consider two: the ESA operational analysis
and the Karlsruhe Institute of Technology Institute of Meteorology and Climate Research (IMK) and the Instituto de Astrofísica
de Andalucía (IAA) analysis, both described in the following subsections.

### 2.3.1 ESA MIPAS

We use MIPAS Level 2 Prototype Processor version 6 (ML2PP v6) of the ESA operational analysis. Early versions of the ESA
MIPAS gas retrievals are described in Raspollini et al. (2006) (full-resolution Instrument Processing Facility version 4.61 (IPF
v4.61)) and the ML2PP v6 upgrades and reduced resolution adaptations are described in Raspollini et al. (2013). Retrievals are
made using a global fitting scheme followed by a posteriori Tikhonov regularization with self-adapting constraints (Raspollini
et al., 2013). The ML2PP v6 data provide retrieved VMR vertical profiles of ten atmospheric gases between approximately 6
to $70\,\mathrm{km}$. Temperature and pressure are retrieved from the spectra at each tangent point of a limb scan and a corresponding
altitude grid is built from the lowest engineering tangent altitude using the equation of hydrostatic equilibrium. Initial guesses
for vertical profiles of a target trace gas, temperature and interfering species are the weighted average of the results from the
previous scan, an appropriate merging of IG2 (initial guess 2) climatological profiles (Remedios et al., 2007), and, if available,
data from the European Centre for Medium-range Weather Forecasts (ECMWF). Spectra are computed using a specialized
linelist derived from HITRAN 1996 (Rothman et al., 1998).

The IPF v4.61 $CH_4$ data product has been validated by Payan et al. (2009) against four balloon instruments, including SPI-
RALE, three aircraft instruments, six ground-based FTSs (all are considered herein), and HALOE. They found good agreement
with a 5 % positive bias in the lower stratosphere and upper troposphere. ML2PP v6 $CH_4$ was compared with BONBON air
sampling measurements by Engel et al. (2016). The reduced-resolution $CH_4$ measurements (2005–2012) agree with in situ data
within 5–10 %. $CH_4$ (and $N_2O$) from ESA MIPAS have been assimilated by the BASCOE code and the assimilated products
have been compared with MLS and ACE-FTS (Errera et al., 2016). The analysis has proven the high quality of the MIPAS data,



but it has also identified the presence of some outliers, especially in the tropical lower stratosphere, and some discontinuities due to issues in the measurements.

### 2.3.2 IMK-IAA MIPAS

The IMK-IAA MIPAS retrieval algorithm has been developed to include and account for deviations from local thermal equilibrium. The data presented here are IMK-IAA reduced-resolution version V5R_CH4_224/225. The early retrieval algorithms are described by von Clarmann et al. (2009), and the updates made to the current version are described by Plieninger et al. (2015). Temperature and tangent altitude are retrieved from the spectra, and pressure is computed from the equation of hydrostatic equilibrium. V5R_CH4_224/225 uses the HITRAN 2008 linelist (Rothman et al., 2009). Temperature a priori profiles are determined from ECMWF analyses and MIPAS engineering information. The IMK-IAA retrieval uses Tikhonov first-order regularization in combination with an all-zero $CH_4$ a priori profile, which serves to smooth the profiles.

Validation of the IMK-IAA MIPAS V5R_CH4_222/223 data has been presented in Laeng et al. (2015). They compare data against ACE-FTS, HALOE, the MkIV balloon FTS, the Solar Occultation For Ice Experiment (SOFIE) on the Aeronomy of Ice in the Mesosphere (AIM) satellite, the SCanning Imaging Absorption spectroMeter for Atmospheric CHartographY (SCIAMACHY) on Envisat, and a cryogenic whole-air sampler (collects gas bottle samples during aircraft flights). They found an agreement within 3 % in the upper stratosphere with other satellite instruments, but in the lower stratosphere, below 25 km, a high bias was found in the MIPAS retrievals of up to 14 %. The V5R_CH4_224/225 has more recently been validated by Plieninger et al. (2016), using ACE-FTS, HALOE, and SCIAMACHY. They found MIPAS $CH_4$ retrievals to be larger by around 0.1 ppmv below 25 km, or around 5 %.

### 2.4 NDACC

NDACC is a global network of a variety of instruments that provides measurements of tropospheric and stratospheric gases that are directly self-comparable (Kurylo and Zander, 2000). The network consists of over 70 stations sparsely distributed at all latitudes. Information about NDACC is available at www.ndacc.org. In this work, we only consider a small subset of NDACC stations that feature high-resolution FTSs and provide a $CH_4$ VMR vertical profile data product via the NDACC data base. Sepúlveda et al. (2012, 2014) demonstrated the good quality of $CH_4$ profiles that can retrieved from the NDACC FTS measurements. The stations are listed in Table 1, along with their locations, references, and some information about each instrument.

The stations do not use identical instruments, spectroscopic lines, or retrieval methods. All but one station use a version of a Bruker 120/5 M or HR, and have predominantly adopted, or upgraded to, the Bruker 125HR. Some stations have more than one instrument, and the type of instrument has changed over time at many of the stations. Toronto, 43.6° N, uses a Bomem DA8.

Retrievals are generally performed using either PROFFIT (Hase et al., 2004) or SFIT4 (Pougatchev et al., 1995) following harmonized retrieval settings recommended by the NDACC IRWG (Sussmann et al., 2011, 2013). Data used herein are from the NDACC database. A summary of retrieval settings is provided by Bader et al. (2016). Lauder and Arrival Heights, at





45.0° S and 77.8° S, use a retrieval strategy that adheres to that defined in Sussmann et al. (2011), with a relaxed Tikhonov regularization constraint at Arrival Heights due to the dynamical nature of the Antarctic atmosphere. Jungfraujoch, at 46.6° N, uses SFIT2. It has been established within the NDACC IRWG that the regularization strength of the $CH_4$ retrieval strategy should be optimized so that the number of degrees of freedom for signal (DOFs) is limited to approximately 2 (Sussmann
et al., 2011).

## 3 Data set variability

To provide context for the VMR differences found when comparing each instrument to TANSO-FTS, shown in Sect. 6, we have examined the internal variability for each instrument. Because the observation geometries and rates of spectral acquisition are different for each instrument, our internal comparisons differ for each instrument. Such differences include the much higher
data density of TANSO-FTS and MIPAS compared to ACE-FTS, which only makes two sets of observations per orbit; and that the NDACC stations are stationary.

Following Holl et al. (2016), we are aware that TANSO-FTS $CH_4$ retrievals are dependent on the a priori used, especially at high altitudes. To examine the variability within the TANSO-FTS data set, we examined the distribution of VMR vertical profiles and of the difference between the retrieval and the a priori, to determine whether the variability of the TANSO-FTS
retrievals matched that of their a priori. We examined 3000 TANSO-FTS measurements by interpolating the a priori and retrieved profiles to the pressure grid used in our comparisons (Sect. 6.1), then computed the difference between the retrieval and the a priori at each pressure level, and their mean and standard deviation. Fig. 1 shows the mean ±1 standard deviation of the difference between the TANSO-FTS $CH_4$ retrievals and their corresponding a priori profiles. The peak value is 0.03 ppmv near 10 km (∼1.5 %) with a standard deviation of the same magnitude.

To examine the variability of the ACE-FTS $CH_4$ data product, we compared each ACE-FTS sunset/sunrise measurement in a year to the sunset/sunrise measurement acquired on the next orbit, taking care to avoid a comparison between sunset and sunrise occultations, or when an acquisition was not made during an orbit. Considering all sunset occultations in 2011, there were 1402 retrieved vertical profiles, and 820 sequential pairs. These pairs are separated by 97 minutes and have a mean spatial separation of 1180 ± 20 km, depending on the latitude of the measurement. For each pair, we computed the difference between
pairs of VMRs on the ACE-FTS 1 km tangent altitude grid, and then found the mean and standard deviation, which are shown in Fig. 1. Within the ACE-FTS data, the largest systematic variability (−0.004 ppmv) occurs around 30 km, with extreme outliers being observed at the lowest tangent altitudes. The mean magnitude of the ACE-FTS variability is 0.002 ppmv (0.1 %) at all altitudes, and 0.009 ppmv below 15 km (0.4 %).

To examine the variability of the MIPAS data sets, we compared the vertical profiles retrieved by IMK-IAA and ESA that are from the same set of MIPAS limb observations and within our coincident data set. For each pair of retrieved vertical profiles from a single set of MIPAS spectra, we interpolated the ESA retrieval to the IMK-IAA 1 km grid and computed their difference (IMK-IAA − ESA), and then found the mean and standard deviation. Fig. 1 shows the mean ±1 standard deviation for this comparison. The two retrievals show good agreement above 30 km (not shown), while the IMK-IAA data has a positive





bias relative to the ESA data product of around 0.15 ppmv between 20 and 30 km. This bias is consistent with the validation results presented in Laeng et al. (2015). The ESA and IMK-IAA comparison exhibits the largest variability, with a mean magnitude (mean of absolute values) of 0.05 ppmv (2 %) for the altitude range considered (9–34 km). Since the two products use the same spectra, it is possible that part of the internal instrument variability is hidden in this approach.

To investigate the variability of the NDACC data, we considered the $CH_4$ VMR vertical profiles in coincidence with TANSO-
FTS and found a subset of NDACC measurements for each site that were made on the same day. We then computed the $CH_4$ VMR differences for each pair of measurements made on the same day (earlier profile minus later; if there are multiple profiles, the differences are all found relative to the earliest). The mean and standard deviation of these differences is also shown in Fig. 1. When examining dates with several measurements, the NDACC differences show a systematic mean increase in tropospheric $CH_4$ with time during a single day. This variability is small, however, with a mean of $-0.004$ ppmv below 30 km and a peak
at 12 km of $-0.006$ ppmv (0.3 %).

Of the four datasets, our variability investigation found that the ACE-FTS data exhibits the smallest variability between measurements, and MIPAS exhibits the largest, while NDACC and TANSO-FTS are of similar magnitudes. The magnitude of the internal variability of the data sets is between 0.05 ppmv (e.g., for NDACC and ACE-FTS in the upper troposphere) and 0.004 ppmv, or around 2 % (e.g., for TANSO-FTS in the upper troposphere and when comparing ESA and IMK-IAA MIPAS).

# 4 Coincidences

Due the coverage and data collection rates of each instrument, different coincidence criteria were used. In the case of ACE-FTS, which only records two occultations per orbit, and NDACC stations, which are stationary, the objective of the coincidence criteria was to maximize the number of measurements used. Conversely, in the case of MIPAS, which makes frequent observations, the objective was to reduce the number of potential coincident measurements. For ACE-FTS and NDACC, we sought mea-
surements made within 12 hours and within 500 km of each TANSO-FTS measurement (spatial separation calculated using the Vincenty method). For the MIPAS data sets, we sought measurements made within 3 hours and 300 km. When searching for MIPAS–TANSO-FTS coincidences within 12 hours and 500 km, we find approximately 180,000 coincidences per month.

TANSO-FTS $CH_4$ VMR vertical profiles tend not to be sensitive above the upper troposphere (see Sect. 5), while ACE-FTS and MIPAS retrievals have a limited vertical extent in the troposphere. To ensure that measurements made by each instrument
overlap, a restriction was placed on ACE-FTS and MIPAS measurements: that their retrieved vertical profiles extend to low enough altitudes, after applying data quality criteria. For ACE-FTS, this requirement was 10 km. For MIPAS, this requirement was relaxed to less than 12 km. IMK-IAA MIPAS $CH_4$ VMR vertical profile retrievals do not extend as low as those made by ESA, to the extent that having the same restriction on altitude range results in only a quarter as many coincidences as the ESA data product. Relaxing the constraint to only 12 km maintains the assurance that retrieved VMRs will overlap with the TANSO-FTS altitude range, though there are only 60 % as many IMK-IAA coincidences compared to ESA coincidences.

TANSO-FTS makes nadir observations in a grid pattern by sweeping its line-of-sight across its ground-track. This results
in a high density of vertical profiles, such that, for a single observation made by ACE-FTS, MIPAS, or NDACC, there are an





average of 11 coincident TANSO-FTS measurements. The subsequent measurement made by MIPAS or an NDACC station will be coincident with a similar number of TANSO-FTS measurements, and most of those will also be coincident with the previous MIPAS or NDACC measurement. A common way to deal with multiple coincidences is to take the mean of the VMR vertical profiles from each instrument, and to compute the difference of the means. When comparing MIPAS to TANSO-FTS,

however, this results in some measurements contributing to the analysis more times than others, biasing the computed VMR difference profiles. Furthermore, this leads to using a mean TANSO-FTS VMR vertical profile that is strongly smoothed, while a coincident ACE-FTS (or NDACC, depending on the station's rate of acquisition at the time) VMR vertical profile is not.

To reduce biases caused by over-counting, when comparing TANSO-FTS to MIPAS, and by smoothing, when comparing TANSO-FTS to ACE-FTS, we reduced the number of coincident measurements by seeking a set of one-to-one coincidences

for unique measurements in the sparser dataset (which is always ACE-FTS, MIPAS, or NDACC). For each measurement that is being compared to TANSO-FTS, we find the TANSO-FTS measurement with the minimum of the sum of ratios of distance in space and time to the coincidence criteria, giving equal weight to both parameters as: $\min(dx/x_{crit} + dt/t_{crit})$, where $dx$ and $dt$ are the distance and time between a given measurement and a TANSO-FTS coincidence, and $x_{crit}$ and $t_{crit}$ are the coincidence criteria. This method is similar to using a $z$ score to compare the spatial and temporal separation, but the sample

size of the set of TANSO-FTS measurements coincident with another measurement is on the order of only ten. Furthermore, the mean and standard deviations of $dx$ and $dt$ reflect the time and distance between each consecutive TANSO-FTS measurement, rather than the time and spatial separation between each TANSO-FTS measurement and those from MIPAS, ACE-FTS, or NDACC.

Table 2 shows the total number of coincidences found between TANSO-FTS and each validation target instrument, as

well as the subsets of unique TANSO-FTS measurements and the one-to-one coincidences used in this paper (equivalent to the number of unique measurements made by each target instrument). Fig. 2 shows an example of the global distribution of coincident measurements. Shown are the first 200 one-to-one coincidences after 1 January 2012. For the ESA and IMK-IAA MIPAS data products, this number of coincidences is found in around two weeks. For ACE-FTS and the NDACC stations (combined), these coincidences occur over several months.

## 30  5  Averaging kernels

The averaging kernels of a profile retrieval provide information about the contributions of the retrieval from a priori information and the measurements. In this study, the retrieval methods for each data set differ and the averaging kernel matrices are differently defined. In general, the rows of the averaging kernel matrix are peaked functions with widths indicative of the vertical resolution of the measurement. The sum of the rows of the matrix gives the sensitivity, or response, of the retrieval. A sensitivity close to one indicates that most of the information in the retrieval comes from the measurement, while sensitivities less than one indicate increased reliance on the a priori in the solution.





The rows of the averaging kernel matrices for the ESA MIPAS, IMK-IAA MIPAS, TANSO-FTS, and the Eureka NDACC station are shown in Fig. 3. Each panel shows the mean from 30 retrievals, with each averaging kernel interpolated to a common pressure grid for that instrument.

In this study, we treat TANSO-FTS retrievals as having the coarser vertical resolution in all cases, despite the widths of the kernel functions shown in Fig. 3a, which are comparable to MIPAS and narrower than NDACC. The peak locations of the

TANSO-FTS averaging kernels do not match the corresponding pressure level of each kernel, therefore the full-width at the half-maximum values when considering the location of the appropriate pressure level are much larger than the full-width at half-maximum values for the averaging kernels of the other instruments.

In the NDACC retrievals, the a priori has a large role and information coming from the measurements can hardly distinguish the contribution coming from the different altitudes. This leads to wide, overlapping averaging kernels. The IMK-IAA MIPAS

retrievals use a form of Tikhonov regularization without an a priori. The ESA MIPAS retrievals use the regularizing Levenberg-Marquardt approach (where the parameter setting has been chosen to leave results largely independent from the initial guess profiles) and a posteriori Tikhonov regularization without an a priori. The ACE-FTS retrievals do not use a regularized matrix inverse method. Consequently, the ACE-FTS and IMK-IAA MIPAS averaging kernels are very narrow, their peak values are close to one at each altitude where a spectrum was acquired, and the solutions do not rely on a priori information. Very similar

averaging kernel are obtained also for ESA MIPAS, with wider widths at lower altitudes where the retrieval grid used is coarser than the measurement grid. The sensitivity of both ACE-FTS and MIPAS is close to one at all altitudes, falling off above 60 or 70 km. This is shown in Fig. 3e. ACE-FTS averaging kernels are under development and preliminary work is shown in Sheese et al. (2016).

The typical sensitivity of an NDACC retrieval is close to unity until above 20 km, falling off towards zero through 60 km.

The sensitivity of TANSO-FTS only reaches 0.2–0.3 between 5–10 km. The implication of such low values for sensitivity is that the TANSO-FTS retrievals are highly dependant on their a priori.

The trace of the averaging kernel matrix gives the DOFs, which are shown for the TANSO-FTS, IMK-IAA MIPAS, ESA MIPAS, and NDACC retrievals are shown in Fig. 4 for retrievals made in the Arctic (for example), above 60° N. The IMK-IAA MIPAS and TANSO-FTS data are in coincidence with one another. The NDACC data come from Eureka, Ny Ålesund, and

Thule. The NDACC and ESA MIPAS data shown are the TANSO-FTS one-to-one coincidences used throughout this study (but are not coincident with the TANSO-FTS data shown in the top panel of Fig. 4). Recreating this figure over mid-latitudes or the tropics reveals a flat trend over time, while over Antarctica, the trends are reversed in DOFs-space.

The mean of the DOFs for the three NDACC stations over the Arctic is 1.98 with a standard deviation, $\sigma$, of 0.50. Over the tropics, considering data from Izaña, La Réunion St. Denis, Altzomoni, and Mauna Loa (La Réunion Maïdo only has data from 2013 onward, not shown here), the mean is 2.39 with $\sigma = 0.37$. The mean DOFs for IMK-IAA MIPAS are slightly larger

than those for ESA MIPAS. Over the Arctic, their means and standard deviations are 17.05, $\sigma = 1.06$ and 15.76, $\sigma = 0.93$, for IMK-IAA and ESA, respectively. Over the tropics, they are 16.10, $\sigma = 0.33$ and 15.88, $\sigma = 1.20$.

The TANSO-FTS DOFs are larger at low latitudes, with a mean over the tropics of 0.72 and $\sigma = 0.08$, and means over the Arctic and Antarctic of 0.32 and 0.20, respectively ($\sigma = 0.13$ and 0.12). The DOFs for a TANSO-FTS retrieval rarely go above



unity. Conversely, in the coincident NDACC data discussed above, over the tropics and Arctic, the DOFs never fall below unity.

Note that the averaging kernel matrices for TANSO-FTS, and therefore the DOFs, cover a much smaller altitude range than for NDACC and MIPAS, which can extend above 100 km.

## 6 VMR vertical profile comparisons

### 6.1 Methodology

Retrievals made by an instrument with fine vertical resolution may result in structure over its vertical range that is not distin-

guishable in retrievals made by an instrument with coarser vertical resolution. In order to make the best comparison between two instruments with differing vertical resolution, it is necessary to smooth the vertical profiles retrieved from the finer resolution instrument, in order to simulate what we could infer from it if it had a similar sensitivity as the other instrument. Smoothing is done using the a priori $CH_4$ VMR vertical profiles and averaging kernel matrices of the instrument with lower vertical resolution (Rodgers and Connor, 2003):

$$\hat{\mathbf{x}}_s = \mathbf{x}_a + \mathbf{A}(\hat{\mathbf{x}} - \mathbf{x}_a),\qquad(1)$$

where $\hat{\mathbf{x}}$ is original higher-resolution retrieved profile, $\hat{\mathbf{x}}_s$ is the smoothed profile, $\mathbf{x}_a$ is the a priori profile of the lower-resolution retrieval, and $\mathbf{A}$ is the averaging kernel matrix of the lower-resolution retrieval ($\mathbf{x}_a$ and $\mathbf{A}$ are from the TANSO-FTS retrieval in all cases presented here). The smoothed profile, $\hat{\mathbf{x}}_s$, approximates the a priori, $\mathbf{x}_a$, when either the rows of $\mathbf{A}$ are close to zero, or the retrieval is close to $\mathbf{x}_a$. As can be inferred from Fig. 3a, above 20–25 km $\hat{\mathbf{x}}_s \sim \mathbf{x}_a$.

In order to apply Eq. 1, all the variables on the right hand side must be interpolated to a common grid. TANSO-FTS retrievals are done on a retrieved pressure grid. Determining the altitude of its VMR vertical profiles requires applying the equation of hydrostatic equilibrium, and incorporating a priori temperature and water vapour. Since pressure is retrieved by ACE-FTS and MIPAS, and the tropospheric a priori pressure profiles and measured surface pressure are accurate for NDACC, all comparisons here have been done on a common pressure grid.

The data products do not always overlap over the entire pressure range of the common grid. For ACE-FTS and MIPAS, we use $\mathbf{x}_a$ to extend their retrieved profiles below their altitude range to cover the full pressure range of the TANSO-FTS averaging kernels. The averaging kernels at these non-overlapping pressure levels do not contribute to the smoothed retrieval at higher, overlapping levels. The following steps are taken to compute vertical profiles of the mean $CH_4$ VMR differences:

1. appropriate instrument data quality flags are applied to each VMR vertical profile in the coincidence pair,

2. TANSO-FTS a priori and validation target VMR vertical profiles are interpolated to the TANSO-FTS retrieval pressure grid,

3. the interpolated validation target profile is extended as needed to match the TANSO-FTS pressure range (and vector length) using the TANSO-FTS a priori,



4. the interpolated validation target profile is smoothed using the TANSO-FTS averaging kernel matrix using Eq. 1,

5. TANSO-FTS retrieved and validation target smoothed VMR vertical profiles are interpolated to a standard pressure grid, levels outside the pressure range of the target's VMR profile are discarded,

6. the piecewise difference between the TANSO-FTS and the smoothed validation target VMR vertical profiles is found,

7. the means, standard deviations, and correlation coefficients of the VMR differences are calculated at each level of the standard pressure grid for all coincidences within a latitude zone.

For comparison, mean VMR vertical profile differences were also computed without smoothing by using only steps 1, 5, 6, and 7. Zonally averaged VMR difference profiles are presented in Sect. 6.2 and results obtained without applying smoothing to the validation targets are shown in Sect. 6.3. The data quality flags in step 1, referring to variables in the data product files, were, for TANSO-FTS: $CH4ProfileQualityFlag$ must be zero; for ACE-FTS: $quality\_flag$ must be zero, and cannot be equal to four, five, or six at any altitude; for ESA MIPAS: $ch4\_vmr\_validity$ must be one and $pressure\_error$ cannot be NaN; for IMK MIPAS: $visibility$ must be one, and $akm\_diagonal$ must be greater than 0.03.

Holl et al. (2016) found that looking for coincident $CH_4$ VMR vertical profile pairs that may have one or both profile locations within a polar vortex, and then filtering these events, had little effect on their vertical profile comparisons below 25 km. Polar vortex event will have a much smaller affect on this study since it uses global and year-round data sets. For these two reasons, our method does not filter for profiles located within a polar vortex. Arrival Heights may be differently affected by a much-stronger Antarctic polar vortex, but comparison results from this site are not anomalous and only accounts for 1.5 % of the NDACC data set so are treated in a consistent manner.

### 6.2 Zonally averaged VMR profile differences

Following Holl et al. (2016), we are trying to determine whether there are any zonal biases in the TANSO-FTS data, or zonal dependencies when making comparisons to other instruments. The mean $CH_4$ VMR differences, averaged zonally, between the TANSO-FTS vertical profiles and the smoothed vertical profiles from ACE-FTS, IMK-IAA MIPAS, ESA MIPAS, and each NDACC station are show in Fig. 5. Each row in Fig. 5 shows the results from five latitudinal zones: 90–60° N, 60–30° N, 30° N–30° S, 30–60° S, and 60–90° S. The left-most column shows the mean differences between the retrievals from TANSO-FTS and those from the other instruments, always calculated as TANSO-FTS − $target$. One standard deviation is shown for each instrument comparison with dotted lines. The middle-left column shows the mean differences as a percentage of the mean $CH_4$ VMR vertical profile taken for the target validation instrument in each zone. The number of VMR measurements used in the mean at each altitude, for each comparison, is shown in the right-most panel, with ESA MIPAS always having the most. At each altitude, we also calculated the Pearson correlation coefficient between the set of TANSO-FTS $CH_4$ VMR measurements and the coincident set from each validation instrument. These are shown in the middle-right column for each panel in Fig. 5.

For each zone, the mean difference tends towards zero and the standard deviation falls off above 100 hPa. This is a reflection of the TANSO-FTS sensitivity. Above this altitude, the TANSO-FTS averaging kernels tend to zero, as shown in Fig. 3, and





the smoothed profiles from each target instrument begin to approximate the TANSO-FTS a priori. Likewise, the TANSO-FTS retrieval above this pressure level is also close to its a priori. Conversely, the number of $CH_4$ VMR measurements in the mean falls off sharply below 10–12 km, or around 80–90 hPa, for the comparisons to the satellite instruments. For the satellite instruments and many of the NDACC stations we see the same trend: a positive bias (TANSO-FTS VMRs are greater than those of the validation instruments) decreasing with increasing altitude, with a tropospheric mean of around 0.02 ppmv, or 1 %. The bias is smallest for the two MIPAS data products in the tropics, between 30° N and 30° S. The bias relative to ACE-FTS is consistent in all the zones. For three of the NDACC stations, Ny Ålesund, Bremen, and Toronto, there is a negative bias (TANSO-FTS retrieves less $CH_4$ than these stations), and for Eureka and Jungfraujoch the bias is close to zero.

There is a notable feature just below 100 hPa in all the zones except 30–60° S. This feature is a pronounced increase in the mean difference in the northern zones 60–30° N and 90–60° N, while it is a decrease in the mean difference between 30° N–30° S, and 60–90° S. It is around this pressure level, or altitude, that the VMR of $CH_4$ begins to fall off rapidly from between 1.8 to 2 ppmv in the troposphere towards 0 ppmv in the upper stratosphere and mesosphere. This feature indicates that the altitude at which this VMR decrease differs between instruments. In the northern hemisphere, this decrease in $CH_4$ VMR occurs at higher altitudes for TANSO-FTS than for the other instruments, and in the tropics and southern hemisphere, this decrease occurs more rapidly and at lower altitudes for TANSO-FTS.

For all instruments and in all zones, the correlation coefficients, $R^2$, at each altitude fall off very sharply, to around 0.2, below 90 hPa (and remain higher in the tropics). This indicates that biases seen in the mean differences are not uniform across the coincident data set and that there is significant variability in the magnitudes of the differences for individual vertical profile pairs, and in the direction of the difference. This is related to the increasing standard deviation of the differences with decreasing altitude, but also to the standard deviations of each data product in the comparison. The sharpness and altitude of the decrease is directly related to the TANSO-FTS averaging kernels. Above 100 hPa, the standard deviations of the TANSO-FTS and the smoothed validation target fall off very sharply as they both begin to approximate the a priori (which also explains why $R^2$ is close to 1).

### 6.3 Impact of smoothing

This study was also performed without applying any smoothing to the vertical profiles of the target validation instruments. These results are shown in Fig. 6, which has the same panels as Fig. 5. The data have not been separated zonally and the plots show means for all latitudes. The 16 NDACC stations have been combined into a single data set.

This study reveals the actual differences one would expect when using the TANSO-FTS data product, and the behaviour of the comparisons at higher altitudes when the validation targets are unaffected by the TANSO-FTS averaging kernels. Without the smoothing applied, the differences show more consistent behaviour over the pressure, or altitude, range shown. While the magnitude of the differences is much greater without smoothing, it is not consistently biased high or low for all the data products at all altitudes. When comparing to the satellite instruments in the upper troposphere, we find that the TANSO-FTS retrieval has greater $CH_4$ VMRs by around 0.05 ppmv, or around 3 %.



For context, a comparison between the ACE-FTS and ESA MIPAS data products, using profiles that were coincident with the same TANSO-FTS observation, is shown in grey. The mean differences between these two data products are smaller than those relative to TANSO-FTS, but have comparable standard deviations, and a slightly smaller correlation, with $R^2 = 0.5$ and 0.6 in the upper troposphere.

The comparison between TANSO-FTS and NDACC extends below the range of ACE-FTS and MIPAS. NDACC and TANSO-FTS agree very well in this region, between $\pm 0.03$ ppmv, or between $\pm 2$ %. In this case, the NDACC stations retrieve more $CH_4$, on average. The low-altitude NDACC and TANSO-FTS data are also more closely linearly correlated, between 50 and 60 %. It should also be noted that the standard deviation of the TANSO-FTS and NDACC differences is also less than those for ACE-FTS and MIPAS at all altitudes.

## 7   Partial column comparisons

### 7.1   Methodology

For each $CH_4$ VMR vertical profile in a pair of coincident measurements, we computed a partial column and compared those from TANSO-FTS to each of the other instruments to investigate how well correlated the derived $CH_4$ abundances are. For consistency, partial columns must be calculated over the same pressure range, as the number of molecules in the column strongly depends on the altitude range (length of the column) of the integral. To determine the pressure range over which to compute partial columns, we considered the TANSO-FTS averaging kernels.

We investigated the sensitivity of the TANSO-FTS retrievals, as defined in Sect. 5 to find an altitude range which minimizes the partial column dependence on a priori information, ensuring our investigation is focused on retrieved information from TANSO-FTS. Fig. 7 shows a two dimensional histogram of the number of TANSO-FTS profiles, for all validation targets combined for two criteria: setting a requirement that the sensitivity must be greater than some threshold, and the resulting number of usable pressure levels in the integral for each profile. We see that the maximum number of usable levels falls off in an approximately linear manner with increasing sensitivity threshold, and that for any sensitivity threshold there will be a large number of TANSO-FTS $CH_4$ VMR vertical profiles that never meet the sensitivity criteria. Increasing the sensitivity cutoff by 0.05 causes approximately 10,000 additional TANSO-FTS vertical profiles, or around 6 % of the total data set combining all validation targets, to fail to meet the requirement at any altitude. The number of usable pressure levels given a restriction on sensitivity is not normally distributed, as can be inferred from the empty area in the upper right of Fig. 7.

For this study, we have selected a sensitivity threshold of 0.2 and require a minimum of three integrable pressure levels. Approximately 23 % of the TANSO-FTS retrievals do not meet these criteria. In such a case, partial columns are still computed using three pressure levels surrounding the level with the maximum sensitivity that are within the range of the target profile (e.g., not below 10 km when comparing to ACE-FTS). Because the overlapping altitude regions for NDACC and TANSO-FTS measurements extend much lower in the atmosphere than for ACE-FTS and MIPAS, the number of TANSO-FTS profiles that do not meet the sensitivity criteria is much smaller for NDACC.





Partial columns are computed as:

$$\text{Column} = \int_{z_1}^{z_2} \frac{P(z)}{kT(z)} \chi(z) dz, \tag{2}$$

where $z_1$ and $z_2$ bound the integration range over altitude $z$, $P$ is pressure, $T$ is temperature, $\chi$ is the $CH_4$ VMR, and $k$ is the Boltzmann constant. For each instrument, $\chi(z)$ is the retrieved quantity, and retrievals were either performed on a pressure grid, or pressure was retrieved simultaneously. We compute partial columns from vertical profiles after step 5 in Sect. 6.1,

so both the TANSO-FTS and the smoothed validation target profiles have the same pressure at each level in the integration. Since TANSO-FTS retrievals do not have an altitude grid, we use that of the coincident measurement, which corresponds to the pressure levels and should be very accurate within the altitude range considered in this study (upper troposphere to lower stratosphere). Thus, we are integrating over the same altitude range for both instruments. Since ACE-FTS and both MIPAS data products include retrieved temperatures, we use their retrieved temperature. For TANSO-FTS and NDACC, we use their

corresponding a priori temperatures.

Several methods of integration were investigated and the results presented in Sect. 7.2 are derived by simple summation of the integrand multiplied by the bin-width of each data point in km. We also used numerical integration techniques, variations of Newton-Coates and Gaussian quadrature formulas. These did not provide significantly different results due the large size of our sample (i.e., our results are statistics found from the Least-squares method, and small differences in the individual partial

columns due to different integration methods do not introduce bias). Since the analytic function being integrated is not well defined, neither is the uncertainty of the derived partial column. Propagating reported retrieval uncertainties of temperature and VMR provides the most appropriate estimate of uncertainty, which is shown in Fig. 8.

### 7.2   Partial column correlation

The computed partial columns from TANSO-FTS are plotted against of those from each validation instrument in Fig. 8.

The panels for ACE-FTS, ESA MIPAS, and IMK-IAA MIPAS contain measurements for all latitudes, and that for NDACC combines results from all 16 stations. Since IMK-IAA retrievals do not extend as low as those of ESA generally, the altitude range of the partial column integral is often smaller than those of the other instruments, resulting in smaller $CH_4$ abundances. Conversely, abundances when comparing to the NDACC stations are the largest.

The Pearson correlation coefficients, $R^2$ are: 0.9986, 0.9968, 0.9965, and 0.9929 for ACE-FTS, ESA MIPAS, IMK-IAA MIPAS, and NDACC, respectively. The slopes of the fitted correlation lines are all close to unity, and a small bias is seen in the $y$-intercept corresponding to between 0.4 % and 2.8 % relative to the mean partial columns of the validation targets, with the greatest corresponding to the NDACC data. Among the individual NDACC stations, those with the largest correlation function intercept are Mauna Loa, Jungfraujoch, Bremen, Izaña, and Zugspitze ($1.2 \times 10^{23}$–$7.5 \times 10^{23}$). TANSO-FTS has a

negative intercept only with respect to two stations: The correlation coefficients for each station are all greater than 0.96, except for Mauna Loa, Izaña, and Maïdo, La Réunion, which all happen to be islands, and for which a large number of coincident TANSO-FTS measurements would have been made over water (see Sect. 8).



Statistics regarding the distribution of the integration ranges over altitude are given in Table 3. This table gives the number of coincident pairs for each validation instrument for which the TANSO-FTS $CH_4$ VMR vertical profile passed the sensitivity

requirements. It also gives the mean and standard deviation of the lower bound of the integral (lower altitude), the width of the interval (highest altitude minus the lowest), and the number of pressure levels used. As expected, the NDACC stations have the widest altitude range, while the IMK-IAA MIPAS retrievals have the smallest. Note that the column in Table 3 showing number of levels used does not correspond to the mode in Fig. 7 since Fig. 7 considers only the TANSO-FTS averaging kernels and does not reflect the lack of available comparison data at lower altitudes.

Repeating the analysis using unsmoothed data from ACE-FTS, ESA and IMK-IAA MIPAS, and NDACC, the spread in the correlation plots increases and the biases observed in the intercepts increase, while the correlation coefficients remain very close to unity. Fig. 9 shows derived partial column correlation plots for each validation target instrument. The intercept, without smoothing is between 2 and 6 %. The correlation coefficient for the MIPAS instruments is reduced to 0.97.

## 8    Discussion

The objective of this study was to quantitatively assess TANSO-FTS $CH_4$ VMR vertical profile retrievals compared with other FTS instruments, and to further investigate whether there were any biases with latitude or other retrieval parameters. As shown in Sect. 6.2, we did not find a significant difference in mean $CH_4$ VMR profile differences between latitudinal zones.

To investigate further, we consider the $CH_4$ VMR differences averaged over altitude for each coincident pair, for each validation instrument. To choose the altitude range over which to find the mean, we use the same sensitivity criteria developed

in Sect. 7.2. The resulting mean differences between TANSO-FTS and ACE-FTS, MIPAS, and NDACC are shown as a function of latitude in Fig. 10. A bias is seen at all latitudes of $0.01330 \pm 0.00006$ ppmv, when combining results from all four validation instruments. There is also a small slope in the data from each hemisphere, decreasing from the poles to the tropics. Linear fit parameters for the combined data set are given in Table 4. This leads to a bias of around 0.004 ppmv in the tropics (0.25 % of a tropical tropospheric VMR value of 1.8–2 ppmv), and of 0.014 ppmv or 0.020 ppmv at the North and South Pole, respectively

(or around 1 %).

We also compared the differences shown in Fig. 10 to TANSO-FTS retrieval parameters: land or sea mask, sunglint flag, incident angle, both along the scan path and GOSAT track path, and observation mode (see Kuze et al., 2009). We found no biases in our coincident TANSO-FTS dataset related to any of these parameters, or whether the observation was made during night or day. The land or sea mask is an indicator of whether the retrieval was made over land, water, or a combination in the field-of-view. In our data set of all one-to-one coincidences between TANSO-FTS and the validation targets, 54.0 % of

TANSO-FTS measurements were made over water, 36.3 % were made over land, and 9.6 % were a mixture. The sunglint flag indicates whether the positions of the sun, satellite, and observation point are related within a predefined range, qualifying the observation as being made in sun-glint mode. In our data set, only 1.6 % of TANSO-FTS measurements are sun-glint observations, and they are all over water and between ±45° latitude. Finally, 54.1 % of TIR observations were made at night.





## 9 Conclusions

We have investigated the sensitivity, and averaging kernels for the TANSO-FTS $CH_4$ TIR vertical profile data product, and done a global comparison with four other FTS data products. Our comparisons showed that the sensitivity of the TANSO-FTS retrieval is relatively low at all altitudes, and that there is a limitation on the useful upper altitude of its data product of below 15 or 20 km. However, this vertical extent is below the lower altitude boundaries (10–15 km) of the other satellite-based data products. In the troposphere, we found good agreement between TANSO-FTS and NDACC, without a bias. The agreement

between these two data sets persisted regardless of whether smoothing was applied to the NDACC data. Therefore, despite the lower sensitivity of the TANSO-FTS data product, it remains an important and unique data set of global tropospheric $CH_4$ measurements.

In the overlapping altitude ranges of the three satellite data products, we found a small, but consistent, positive bias of around 0.02 ppmv, or 1 %. We found that the shape of the profile near 15 km, where the $CH_4$ VMR vertical profiles fall off

with increasing altitude, does not match that of the other instruments, and in a consistent manner, resulting in a pronounced feature just below the 100 hPa level. Despite the large variability in each data set and in the differences between the TANSO-FTS retrievals and the others, we found that partial columns computed from the vertical profiles were very tightly correlated, with and without smoothing.

When looking for a relationship between latitude and the differences between data products, we found a small, but statisti-

cally significant, dependence of the mean differences, taken over altitude and latitude. The TANSO-FTS data product shows better agreement over the tropics than the poles. We look forward to future versions of the retrieval which may feature a greater sensitivity and altitude range, while reducing the small biases and dependence on the a priori profiles.

*Author contributions.* The work presented here was done by K. S. Olsen with input from, and under the supervision of K. Strong, K. A. Walker, and N. Saitoh. K. S. Olsen prepared the manuscript with contributions from all co-authors. $CH_4$ retrievals were developed

and provided, with additional input, by: N. Saitoh for TANSO-FTS, C. D. Boone for ACE-FTS, P. Raspollini for ESA MIPAS, J. Plieninger for IMK-IAA MIPAS, M. Grutter for Altzomoni, J. W. Hannigan for Thule and Mauna Loa, F. Hase for Kiruna, N. Jones for Wollongong, W. Bader for Jungfraujoch, M. de Mazière for La Réunion St. Denis and Maïdo, J. Notholt for Bremen and Ny Ålesund, M. Schneider for Izaña, D. Smale for Lauder and Arrival Heights, S. Conway and K. Strong for Eureka and Toronto, and R. Sussmann for Zugspitze.

*Acknowledgements.* This research was conducted under the framework of the Japan Aerospace Exploration Agency (JAXA), National Institute for Environmental Studies (NIES), and the Ministry of the Environment (MOE) Research Announcement (RA) project "GOSAT Validation Using Eureka and Toronto Ground-Based Measurements and ACE, CloudSat, and CALIPSO Satellite Data" with Kimberly Strong (Principal Investigator, University of Toronto), David Hudak (Co-Investigator, Environment & Climate Change Canada (ECCC)),

N. T. O'Neill (Co-Investigator, Université de Sherbrooke), and K. A. Walker (Co-Investigator, University of Toronto). This GOSAT RA project was supported by the Canadian Space Agency (CSA), the Natural Sciences and Engineering Research Council of Canada (NSERC), and ECCC. This work is the result of many, long-lasting collaborations, and we would like to thank our co-authors and collaborators for





providing data and expertise. The GOSAT team provided early access to its TANSO-FTS TIR $CH_4$ VMR vertical profiles and regular access to its other data products through the GOSAT User Interface Gateway: data.gosat.nies.go.jp/. SCISAT/ACE is a Canadian-led mission

mainly supported by the CSA and NSERC. The ACE-FTS Science Team at the University of Waterloo provided access to their level 2 data through database.scisat.ca/level2/ace_v3.5, and expert knowledge with its interpretation and quality management. Access to the ESA MIPAS level 2 data was granted and provided through the ESA Earth Online portal: earth.esa.int. The IMK-IAA MIPAS level 2 data were accessed using KIT website: www.imk-asf.kit.edu/english/308.php. NDACC data has been compiled by many independent research groups, and was accessed through a National Centers for Environmental Prediction FTP server, with each station being accessible through:

www.ndsc.ncep.noaa.gov/data/data_tbl/. Measurements at PEARL were made by the Canadian Network for the Detection of Atmospheric Change (CANDAC), led by J. R. Drummond, and in part by the Canadian Arctic ACE/OSIRIS Validation Campaigns, led by K. A. Walker. Support is provided by AIF/NSRIT, CFI, CFCAS, CSA, EC, GOC-IPY, NSERC, NSTP, OIT, PCSP and ORF. Logistical and operational support is provided by PEARL Site Manager P. Fogal, CANDAC operators, and the ECCC Weather Station. Measurements at the University of Toronto Atmospheric Observatory were supported by CFCAS, ABB Bomem, CFI, CSA, EC, NSERC, ORDCF, PREA, and the University

of Toronto. NDACC data analysis at Toronto and Eureka was supported by the CAFTON project, funded by the CSA's FAST Program. The National Institute of Water and Atmospheric Research Ltd (NIWA) operated ground based FTSs are core-funded through New Zealand's Ministry of Business, Innovation and Employment. We thank Antarctica, New Zealand and the Scott Base staff for providing logistical support at Arrival Heights. Measurements at the Jungfraujoch station are primarily supported by the Fonds de la Recherche Scientifique (F.R.S.–FNRS) and the Fédération Wallonie-Bruxelles, both in Brussels. The Swiss GAW-CH program of MeteoSwiss is further acknowl-

edged. We thank the International Foundation High Altitude Research Stations Jungfraujoch and Gornergrat (HFSJG, Bern) for supporting the facilities needed to perform the observations and the many colleagues who contributed to FTS data acquisition at that site. W. Bader has received funding from the European Union's Horizon 2020 research and innovation programme under the Marie Sklodowska-Curie grant agreement No 704951. We would like to thank A. Bezanilla, who operates the Altzomoni site, and W. Stremme who does the data processing for the Altzomoni site and uploads the data to the NDACCW archive. Altzomoni is supported by Consejo Nacional de Ciencia y Tecnología

(CONACYT, grants 239618 & 249374) and la Dirección General de Asuntos del Personal Académico de la Universidad Nacional Autónoma de México (DGAPA-UNAM, grants IN109914 & IA101814).



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

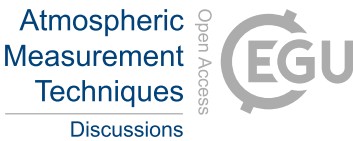

**Table 1.** FTS instruments used in the $CH_4$ VMR vertical profile comparisons presented herein.

| Instrument | Spectral Resolution[a] | Spectral Range[b] | Viewing Geometry | NDACC Latitude | NDACC Longitude | Reference |
|---|---|---|---|---|---|---|
| TANSO-FTS | $0.2\,\mathrm{cm}^{-1}$ | $700$–$1800\,\mathrm{cm}^{-1}$ | nadir | | | Kuze et al. (2009) |
| MIPAS | $0.0625\,\mathrm{cm}^{-1}$ | $685$–$2410^{c}\,\mathrm{cm}^{-1}$ | limb | | | Fischer et al. (2008) |
| ACE-FTS | $0.02\,\mathrm{cm}^{-1}$ | $750$–$4400\,\mathrm{cm}^{-1}$ | solar occultation | | | Bernath et al. (2005) |
| Eureka | $0.0024\,\mathrm{cm}^{-1}$ | $450$–$4800\,\mathrm{cm}^{-1}$ | ground | $80.1°$ N | $86.4°$ W | Batchelor et al. (2009) |
| Ny Ålesund | $0.0015\,\mathrm{cm}^{-1}$ | $475$–$4500\,\mathrm{cm}^{-1}$ | ground | $78.9°$ N | $11.9°$ E | Notholt et al. (1997) |
| Thule | $0.004\,\mathrm{cm}^{-1}$ | $700$–$5000\,\mathrm{cm}^{-1}$ | ground | $76.5°$ N | $68.8°$ W | Goldman et al. (1999) |
| Kiruna | $0.0024\,\mathrm{cm}^{-1}$ | $450$–$4800\,\mathrm{cm}^{-1}$ | ground | $67.8°$ N | $20.4°$ E | Blumenstock et al. (2006) |
| Bremen | $0.0024\,\mathrm{cm}^{-1}$ | $450$–$4800\,\mathrm{cm}^{-1}$ | ground | $53.1°$ N | $8.8°$ E | Buchwitz et al. (2007) |
| Zugspitze | $0.0015\,\mathrm{cm}^{-1}$ | $475$–$4500\,\mathrm{cm}^{-1}$ | ground | $47.4°$ N | $11.0°$ E | Sussmann and Schäfer (1997) |
| Jungfraujoch | $0.0015\,\mathrm{cm}^{-1}$ | $475$–$4500\,\mathrm{cm}^{-1}$ | ground | $46.6°$ N | $8.0°$ E | Zander et al. (2008) |
| Toronto | $0.004\,\mathrm{cm}^{-1}$ | $750$–$8500\,\mathrm{cm}^{-1}$ | ground | $43.6°$ N | $79.4°$ W | Wiacek et al. (2007) |
| Izaña | $0.0024\,\mathrm{cm}^{-1}$ | $450$–$4800\,\mathrm{cm}^{-1}$ | ground | $28.3°$ N | $16.5°$ W | Schneider et al. (2005) |
| Mauna Loa | $0.0024\,\mathrm{cm}^{-1}$ | $450$–$4800\,\mathrm{cm}^{-1}$ | ground | $19.5°$ N | $155.6°$ W | Hannigan et al. (2009) |
| Altzomoni | $0.0024\,\mathrm{cm}^{-1}$ | $450$–$4800\,\mathrm{cm}^{-1}$ | ground | $19.1°$ N | $98.7°$ W | Baylon et al. (2014) |
| St. Denis, La Réunion | $0.0036\,\mathrm{cm}^{-1}$ | $600$–$4300\,\mathrm{cm}^{-1}$ | ground | $20.9°$ S | $55.5°$ E | Senten et al. (2008) |
| Maïdo, La Réunion | $0.0024\,\mathrm{cm}^{-1}$ | $600$–$4500\,\mathrm{cm}^{-1}$ | ground | $21.1°$ S | $55.4°$ E | Baray et al. (2013) |
| Wollongong | $0.0024\,\mathrm{cm}^{-1}$ | $450$–$4800\,\mathrm{cm}^{-1}$ | ground | $34.4°$ S | $150.9°$ E | Kohlhepp et al. (2012) |
| Lauder | $0.0035\,\mathrm{cm}^{-1}$ | $700$–$4500\,\mathrm{cm}^{-1}$ | ground | $45.0°$ S | $169.7°$ E | Bader et al. (2016) |
| Arrival Heights | $0.0035\,\mathrm{cm}^{-1}$ | $750$–$4500\,\mathrm{cm}^{-1}$ | ground | $77.8°$ S | $166.6°$ E | Wood et al. (2002) |

[a] For NDACC instruments, the best achievable spectral resolution is listed here. Operationally achieved spectral resolutions for NDACC instruments are often coarser.

[b] NDACC instruments use optical filters that reduce the effective spectral range when making measurements.

[c] MIPAS' spectral resolution is divided into four, narrower bands.



**Table 2.** Number of coincident $CH_4$ VMR vertical profile measurements that were found between TANSO-FTS retrievals and those from ESA MIPAS, IMK-IAA MIPAS, ACE-FTS, and NDACC stations. The three columns show the total number of coincidences found, the number of unique TANSO-FTS measurements within those coincidences, and the size of the reduced, one-to-one coincidences used.

| Target Instrument | Total Coincident Profiles | Unique TANSO-FTS Profiles | One-to-one Profiles Used |
|---|---|---|---|
| ESA MIPAS | 450,230 | 358,267 | 85,386 |
| IMK-IAA MIPAS | 267,065 | 210,573 | 51,099 |
| ACE-FTS | 51,937 | 47,560 | 4,302 |
| Total NDACC | 213,181 | 44,920 | 17,637 |
| Eureka | 11,843 | 2,447 | 1,009 |
| Ny Ålesund | 5,445 | 1,300 | 349 |
| Thule | 6,997 | 3,359 | 513 |
| Kiruna | 4,595 | 2,056 | 529 |
| Bremen | 2,610 | 1,452 | 211 |
| Zugspitze | 47,512 | 5,743 | 3,469 |
| Jungfraujoch | 18,757 | 5,938 | 1,493 |
| Toronto | 9,909 | 5,195 | 816 |
| Izaña | 56,254 | 4,336 | 4,501[a] |
| Mauna Loa | 4,338 | 2,381 | 379 |
| Altzomoni | 4,746 | 854 | 486 |
| St. Denis, La Réunion | 12,270 | 3,161 | 1,507 |
| Maïdo, La Réunion | 3,139 | 868 | 383 |
| Wollongong | 27,781 | 4,808 | 2,365 |
| Lauder | 7,083 | 2,638 | 704 |
| Arrival Heights | 5,042 | 3,122 | 258 |

[a] The Izaña NDACC coincidence dataset is the only one in which TANSO-FTS
measurements are more sparse. For consistency, Izaña was not treated as a special case.




**Table 3.** Statistics for the partial column integration ranges for ESA MIPAS, IMK-IAA MIPAS, ACE-FTS and NDACC stations with the requirements that the TANSO-FTS sensitivity, $s$, is greater than 0.2 for at least three pressure levels. The number of coincident profiles passing this criterion, $N$, and its percentage of one-to-one coincidences found in this study are given. Means and standard deviations are given for the minimum altitudes, $\min(z)$, total integration range, $z_{range}$, and number of levels used, $n$.

| Target | Profiles with $s > 0.2$ | | Lowest Altitude (km) | | Altitude Range (km) | | Number of Levels | |
|---|---|---|---|---|---|---|---|---|
| Instrument | $N$ | (%) | $\min(z)$ | $\sigma_{\min(z)}$ | $z_{range}$ | $\sigma_{z_{range}}$ | $n$ | $\sigma_n$ |
| ESA MIPAS | 52,016 | 60.9 | 8.4 | 1.5 | 4.6 | 1.5 | 4.8 | 1.1 |
| IMK-IAA MIPAS | 17,787 | 34.8 | 11.3 | 0.6 | 3.5 | 0.9 | 3.7 | 0.6 |
| ACE-FTS | 2,562 | 59.6 | 7.3 | 1.4 | 5.2 | 2.3 | 5.4 | 1.8 |
| Total NDACC | 18,587 | 98.0 | 3.3 | 1.0 | 11.3 | 2.1 | 10.4 | 1.5 |

**Table 4.** Least squares regression statistics for the data in each hemisphere plotted in Fig. 10. Results from all four validation target datasets are combined.

| | Slope (ppmv/° latitude) | Intercept (ppmv) | $R^2$ |
|---|---|---|---|
| Northern | $0.000113 \pm 0.000005$ | $0.0053 \pm 0.0003$ | 0.08 |
| Southern | $-0.000207 \pm 0.000004$ | $0.0031 \pm 0.0002$ | 0.18 |





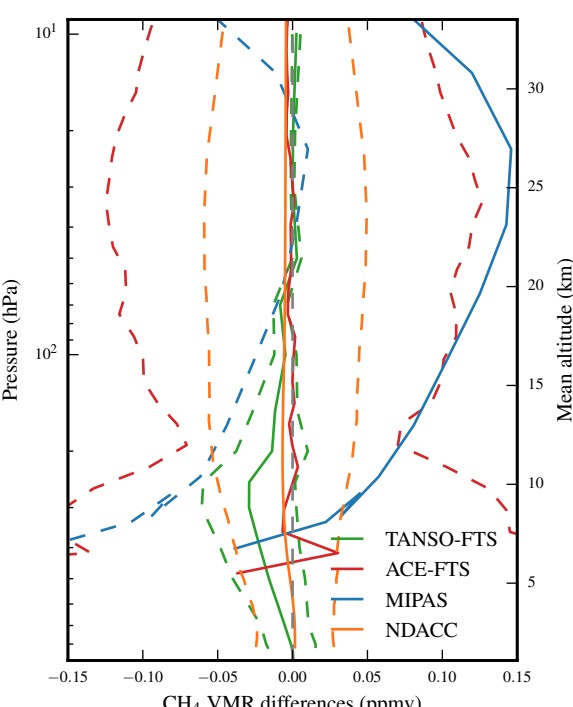

**Figure 1.** Results for investigating the internal variability of each $CH_4$ VMR profile data set, comparing: TANSO-FTS measurements with their a priori (green), sequential ACE-FTS measurements (red), ESA MIPAS measurements with IMK-IAA measurements (blue), and NDACC measurements with others made on the same day (orange). Dashed lines are one standard deviation.



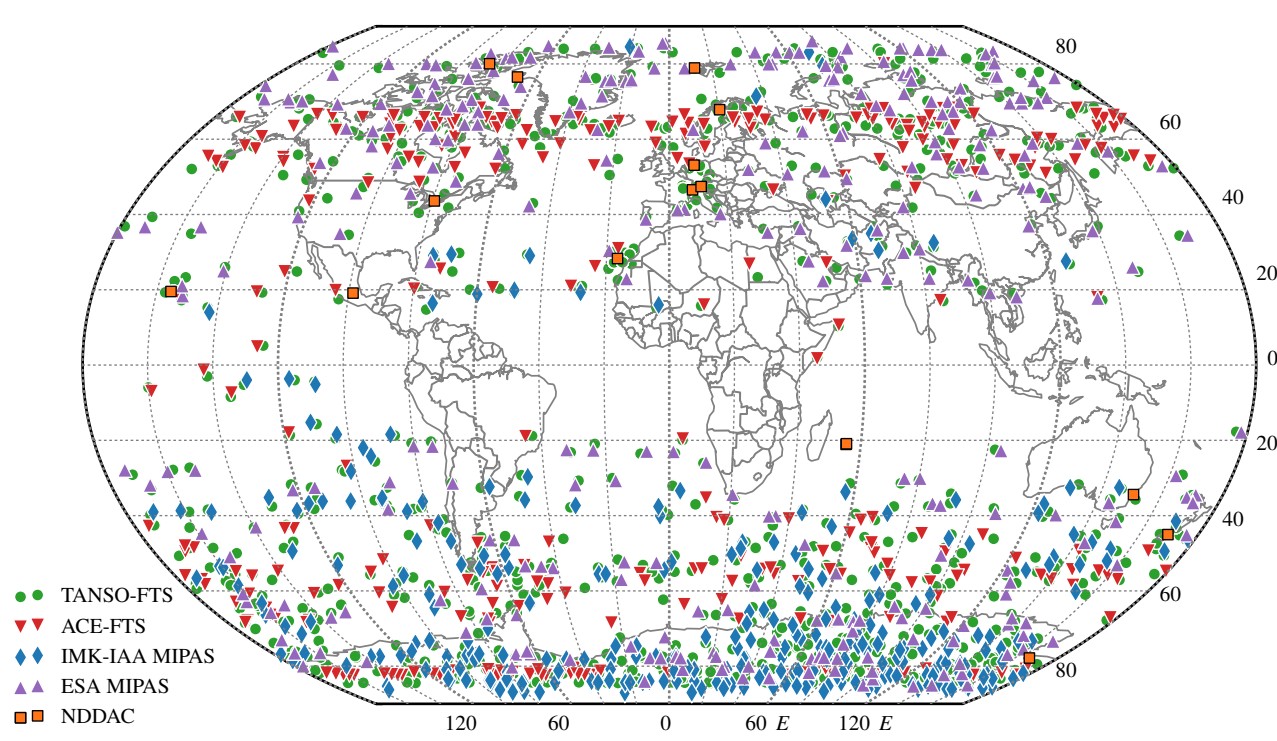

**Figure 2.** Locations of the first 200 observations of 2012 used in this study for TANSO-FTS (green), ACE-FTS (red), and IMK-IAA MIPAS (blue), ESA MIPAS (purple). The NDACC stations are shown in orange.





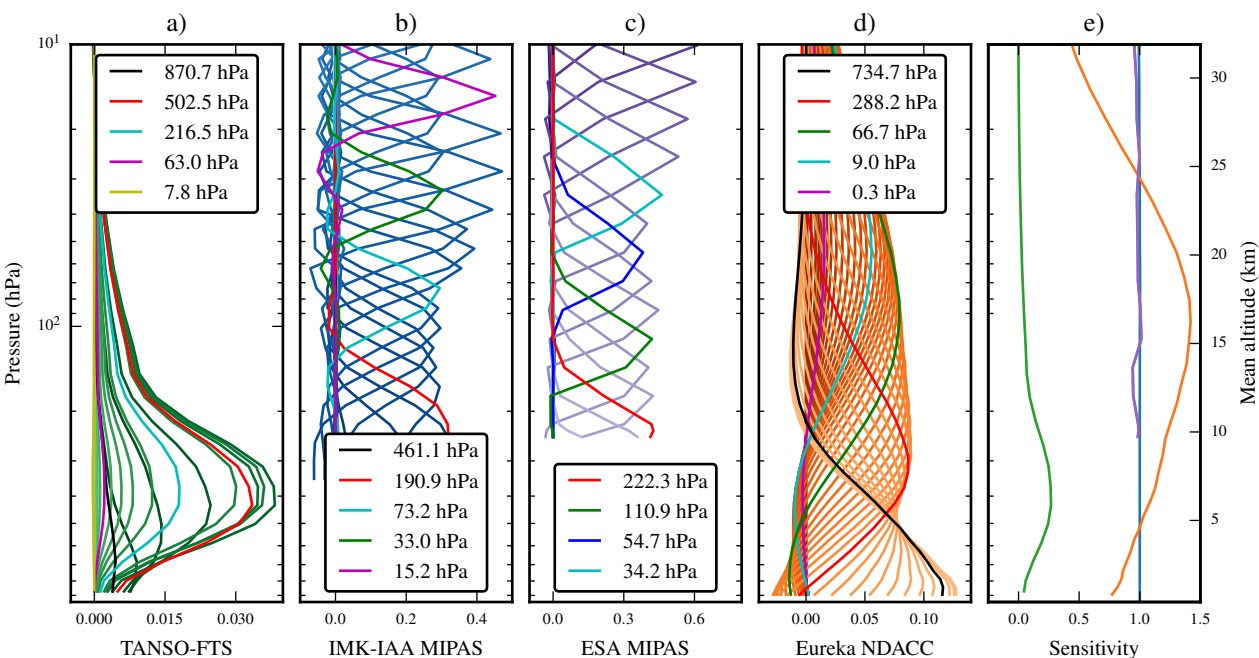

**Figure 3.** Example of averaging kernels for: a) TANSO-FTS, b) IMK-IAA MIPAS, c) ESA MIPAS, and d) NDACC. Each kernel shown is the mean from 30 averaging kernel matrices from measurements made over the Arctic, interpolated to a common pressure grid. Panel d) shows the mean averaging kernels from the Eureka station. Panel e) shows the sensitivity for the mean averaging kernels shown in each panel: TANSO-FTS (green), IMK-IAA MIPAS (blue), ESA MIPAS (purple), and NDACC (orange).

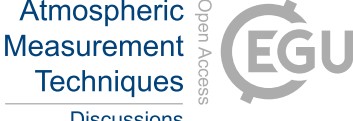



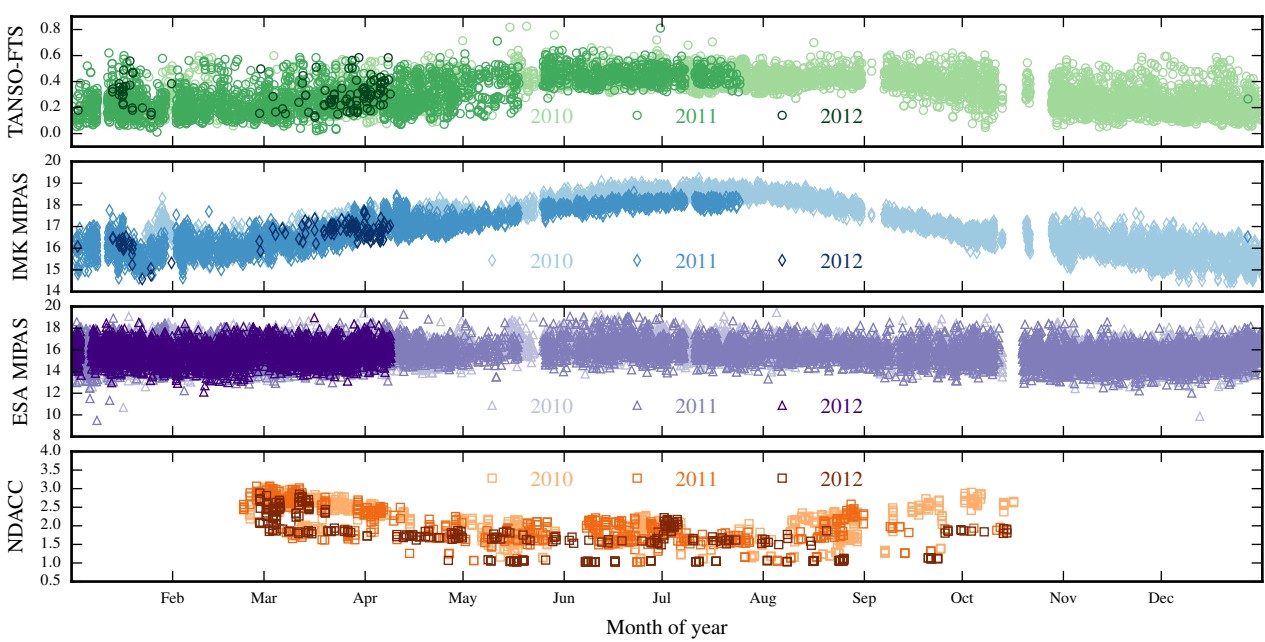

**Figure 4.** Degrees of freedom for signal for, from top to bottom: TANSO-FTS, IMK-IAA MIPAS, ESA MIPAS, and NDACC. Each satellite (and panel) uses a different symbol and colour, but the colour shades indicate the year the measurement was made in. The TANSO-FTS and IMK-IAA MIPAS measurements shown are in coincidence. The ESA MIPAS and NDACC data are from our analyzed data set, but not in coincidence with the TANSO-FTS data in the top panel. All data are from the Arctic, 90–60° N, with the NDACC measurements from Eureka, Ny Ålesund and Thule.





**Figure 5.** Zonally averaged comparison results. The rows present results for each zone, from top to bottom: 90–60° N, 60–30° N, 30° N–30° S, 30–60° S, and 60–90° S. In each row, the four panels show, from left to right, the mean CH$_4$ VMR difference between retrievals from TANSO-FTS and the validation target at each pressure level; the mean CH$_4$ VMR differences relative to the mean CH$_4$ VMR vertical profile of the validation target; the correlation coefficients $R^2$ of the CH$_4$ VMR differences for each coincident pair at each pressure level; and the number of coincidences at each pressure level. Differences are calculated as TANSO-FTS − $target$ for each dataset compared. In all frames, ACE-FTS is shown in red, ESA MIPAS is purple, IMK-IAA MIPAS is blue, and NDACC stations are shades of orange. Each individual NDACC station with a zone is shown, and their shades indicated.





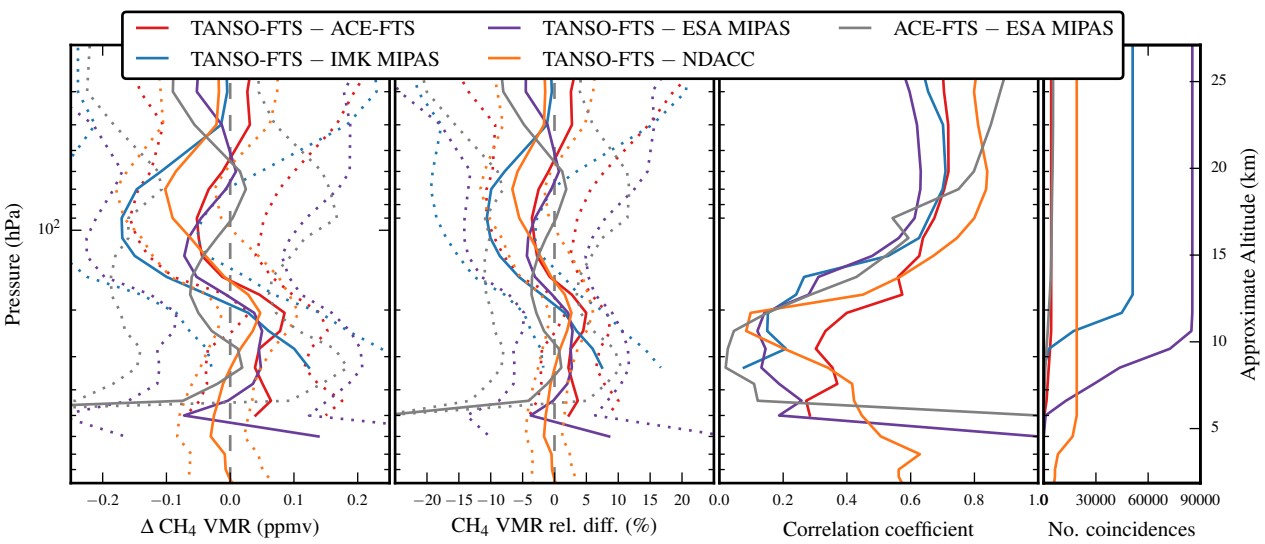

**Figure 6.** Averaged comparison results, as in each panel of Fig. 5, for all latitudes, without applying smoothing to the validation instruments'
$CH_4$ VMR vertical profiles. Differences are calculated as TANSO-FTS $-\ target$ for each dataset compared (and ACE-FTS $-$ ESA MIPAS
for that case).





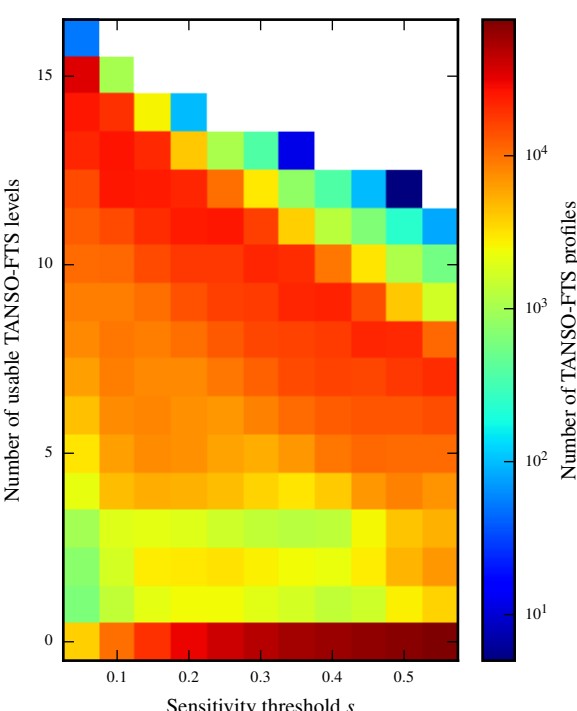

**Figure 7.** Two-dimensional histogram showing the number of TANSO-FTS $CH_4$ VMR profiles within our data set ($z$-axis) that have some number of usable pressure levels ($y$-axis) with a sensitivity greater than some given threshold, $s$ ($x$-axis). The data set shown here consists of all TANSO-FTS observations that are one-to-one coincident with a target validation dataset. The threshold chosen for this study was $s = 0.2$.




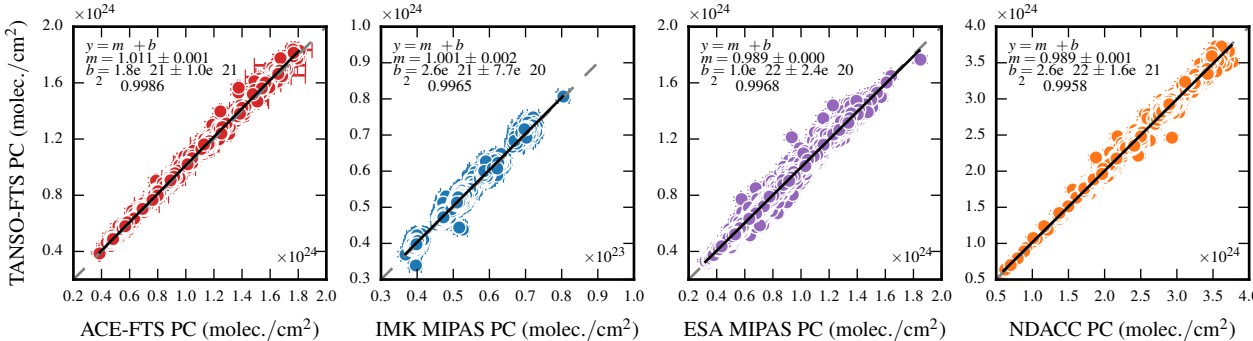

**Figure 8.** Partial column (PC) correlation plots comparing TANSO-FTS $CH_4$ to each validation instrument. Comparisons to ACE-FTS are red, to IMK-IAA MIPAS are blue, to ESA MIPAS are purple, and to NDACC are orange. The vertical range of partial column integration varies for each pair of coincident profiles based on the criteria described in Sect.7.1. The statistics for weighted linear least-squares regression are shown, with weights equal to $1/(\delta_x^2 + \delta_y^2)$.

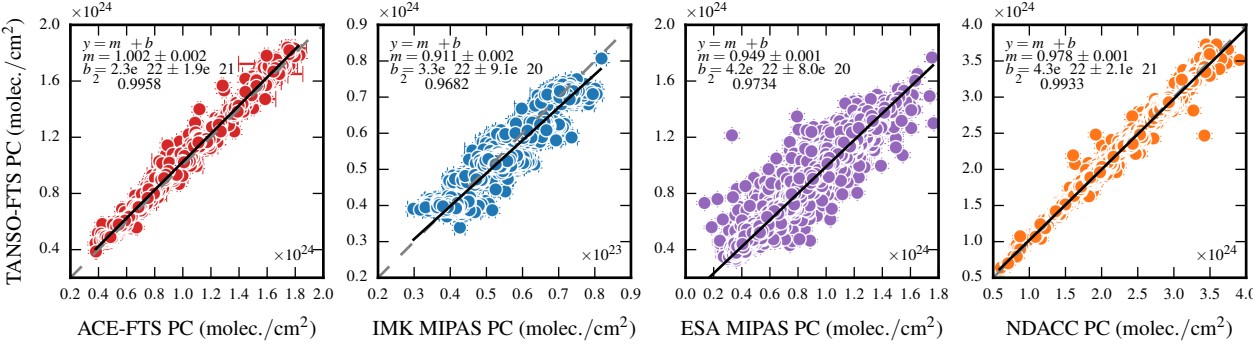

**Figure 9.** As in Fig. 8, but for partial column correlation results using unsmoothed $CH_4$ VMR vertical profiles for each validation instrument.





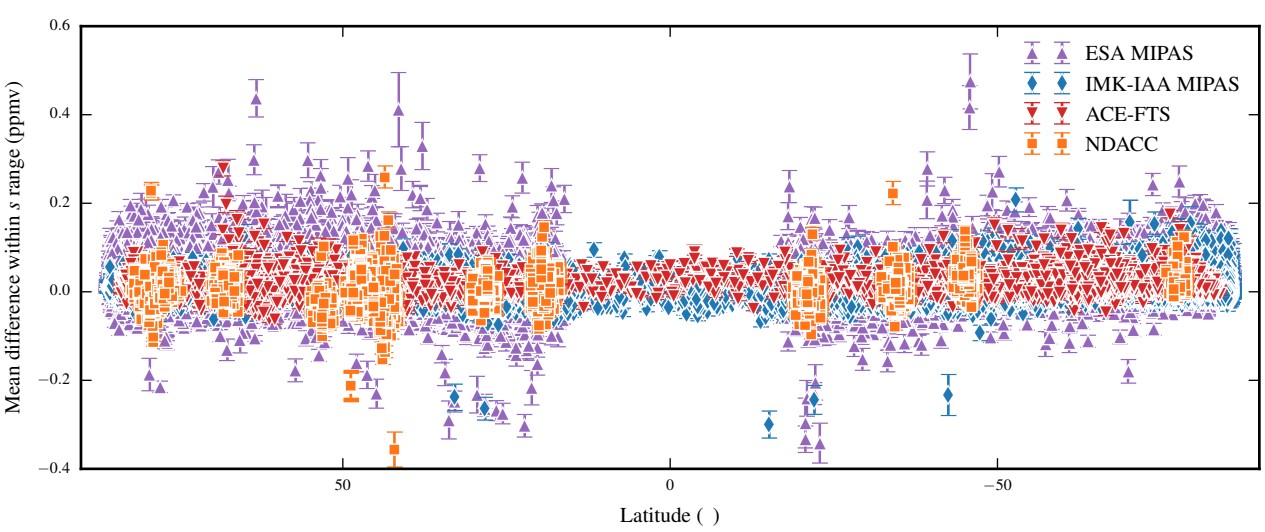

**Figure 10.** Mean $CH_4$ VMR differences between TANSO-FTS and each validation target dataset, averaged vertically using the altitude range selected for integrating partial columns as a function of latitude. Differences are calculated as TANSO-FTS $- target$ for each dataset compared.