# Peer review of "Comparison of the GOSAT TANSO-FTS TIR CH4 volume mixing ratio vertical profiles with those measured by ACE-FTS, ESA MIPAS, IMK-IAA MIPAS, and 16 NDACC stations"

_Atmospheric Measurement Techniques, 2017_

## Referee Comment (RC1) · Anonymous Referee #2 · 28 Apr 2017

Olsen et al. (2017) compares GOSAT TANSO-FTS profile retrievals and partial columns of methane (CH$_4$) to coincident data from ACE-FTS, ESA MIPAS, IMK-IAA MI-PAS, and NDACC, focusing on the upper troposphere/lower stratosphere (UTLS) over 2009-2013. This work expands an earlier TANSO inter-comparison in the Arctic to include global measurements from additional satellite and ground-based remote sensing instruments, aiming to identify possible zonal variability in the profile retrievals. Given the importance of understanding global CH$_4$ and the uncertainty in its recent trends, precise and accurate measurements with global coverage are very much needed. As GOSAT is one of the primary satellites in monitoring atmospheric CH$_4$ today, the assessment of the spatial biases of TANSO measurements would give confidence to the scientific community's use of an important data set.

The approach described in the paper is comprehensive, and the methodology is thought-through. However, the exposition of the significance, conclusions, and limitations of this work require more development. The paper would benefit from a re-balancing of its structure, with the most critical changes being: (1) augmenting the discussion of previous and/or similar validation efforts of TANSO CH4, (2) paring down of the instrument background sections to focus on details directly relevant to their results, and (3) discussing the reasons for and implications of the differences highlighted in the comparison. Therefore, I would recommend publication of this manuscript after these points are addressed.

**1 Scientific Evaluation**

**General Comments**

1. This paper would benefit from a more rigorous analysis (or if already done, a more comprehensive description) of the causes of the differences noted in the results. The authors are thorough in considering different parameters, but the text lacks a synthesis of how these difference relate to the results of the comparisons. For example, what component of the differences between TANSO and the other instruments can be explained by spectroscopy verses a priori profiles used? This can be addressed for the specific parameters already mentioned in the text; e.g. many of the retrievals incorporate the same linelists–do those profiles show better agreement to each other? The implications of these differences for the scientific community's use of the TANSO product based on these differences would also strengthen the paper.

2. At several points throughout the paper, principally in Sections 2-5, the authors include technical details about the satellites/instruments that do not seem pertinent to the study presented (e.g. the inclination angles). The extraneous information was distracting and diluted the narrative of the instrument comparison. I recommend including only those details that the reader should know to understand and evaluate the results and conclusions of this comparison, especially if those details are published elsewhere. For necessary details where the connection to the present study is not clear, explicitly listing the relevance and/or implications would be useful. (e.g. The authors list measurement windows and spectral resolution but need to comment on how these inter-instrument differences might relate to the results.) The authors might also consider moving some of these details that are useful but not central to the paper into an appendix or supplement.

3. Given that methane is generally provided in units of ppb, do the authors have a specific reason for using ppm? If not, I suggest changing the references and figures to ppb. Figure 5 in particular would be more clear without extraneous zeros and decimal places.

**Page 2**

l. 11-13: The redundancy in the list of GOSAT objectives can be pared down. In addition, these objectives should be related, at least in part, to the objectives of this research. i.e. How does this paper contribute to the objectives of the satellites and the scientific community? This question is briefly touched upon on l. 31, but needs to be developed.

l. 24-26: Given the focus on zonal dependence, providing the latitude of Eureka in the text would be useful. Also, please include citations for PEARL and NDACC.

l. 29-31: As the main objective (and contribution) of this paper is to expand TANSO validation globally, more consideration of the issues of spatial coverage is needed, including a literature review of zonal biases (possibly does not exist for this particular TANSO product, but if that is the case this should be stated) and a description of mechanisms

that make the Arctic non-representative (e.g. polar vortex changing vertical profiles of trace gases and reducing the accuracy of a priori information).

**Page 3**

l. 5/Table 1: Please add year ranges to Table 1. It is not clear when mentioning the 2009-2013 time frame in the abstract whether all of the instrument measure consistently over that time period.

l. 7-14: If this is an outline, put section numbers after each sentence. If this is an overview, this paragraph might fit better in the methods section rather than the introduction.

l. 27: What type of coverage? Spatial? Spectral?

l. 28: Is this paper the first time methodology for $CO_2$ is applied to $CH_4$? If so, the retrieval (or the at least the aspects that differentiate it from other TANSO retrievals) should be described more.

l. 33: Do the 2011 updates have a reference (e.g. on the HITRAN website or used in a validation paper)? If so, please include a citation.

**Page 4**

l. 4-6: This sentence, with the important conclusions of the referenced paper added, should be moved to the Introduction, at the end of the second paragraph.

l. 9-11: If information about MAESTRO is relevant, include a reference for the instrument; otherwise this sentence can be removed.

l. 16: Is 5km the lowest altitude for ACE-FTS measurements filtered using the recommended flags (e.g. not a priori values)? Listing the lowest altitude of the data used in this paper would be more relevant, particularly for the discussion on vertical range in subsequent sections.

l. 33 and 36: Do the percentages listed apply to all trace gases or just methane? Please make this more clear.

**Page 5**

l. 4-5: Because this is the data version used, the results from Waymark et al. (2013)

should be summarized, as is done with the above papers.

l. 25: By "initial guesses" do you mean a priori profiles? Please clarify.

**Page 6**

l. 6-7: What are the reasons for the outliers/discontinuities? Would these impacts the results of this study?

l. 30: Using "some information" is vague and does not tell the reader the relevance of the included information.

l. 32-35: Is there a reference for an inter-comparison of these instruments? How are inter-site differences due to different instrumentation accounted for in this study?

l. 36: Are those references for the most recent versions of the retrieval software? The spectroscopy has presumably changed since 1995/2004.

l. 37: Does "harmonized" mean consistent between sites? Please use a more clear term.

**Page 7**

l. 10: Given the differences delineated in this section, have you done a covariance analysis or sensitivity test to assess whether the results of the comparison depend on the retrieval software used, instrument, or any other difference across NDACC sites?

l. 20: Are these measurements representative of year, season, location, tropopause height, etc.?

l. 25: Please define "sunset/sunrise measurement" in this context.

**Page 8**

l. 5-6: Please summarize the results that the bias is consistent with.

l. 24: Why was reducing coincident measurements an objective, if you could average them and thereby reduce potential bias (c.f. Kulawik et al. 2016)? Was this a data processing issue from the large number of coincidences? If the coincidence criteria varies by instrument, some sort of bootstrap or sensitivity test with a subset of data should be run to see if the VMRs are different with the more lax coincident criteria.

**Page 9**

l. 8-9: Including references for the approach mentioned would be useful.

l. 19: Please define a z score and/or include a reference.

l. 26-27/Figure 2: Does using only the first 200 observations of the year capture any time-varying spatial coverage of the satellite data? Would it be more appropriate to use the first 20 observations of each month of 2012, for instance?

**Page 10**

l. 5-6/Figure 3: Please provide a more clear description of what the pressure levels in the legends correspond to (i.e. the pressure widths of the averaging kernel rows vs. the pressure on the y-axis), as relates to the findings of the paper. (You could perhaps include the simple averaging kernel equation if useful, but if the text becomes too detailed I would suggest moving this description to an appendix/supplement.) Also, if these pressure levels are meant to be compared across instruments, using a single colour scale for panels a-d would be helpful, e.g. binned into ranges or following a colour gradient.

l. 8-12: Please add the implications for these comparisons. Related to the previous comment, the remarks on "full-width at the half-maximum values" would be more un-derstandable by rewording the phrase "values when considering the location of the appropriate pressure level" and adding a more clear description of the averaging ker-nel widths (p. 9 l. 33-34).

l. 25-26: How do you determine the influence of this dependence on the results? (This paragraph might need to be moved closer to the discussion of TANSO priors later in the manuscript.)

l. 32: What is the implication of the flat trends over mid-latitudes and tropics, as relates to the objective of this paper to determine zonal dependence of the retrievals?

**Page 11** l. 28: Please include a reference for the claim that the NDACC a pri-ori/measured pressure profiles are accurate. (This might fit better in Section 2.) l. 27-29: I'm not sure I follow the logic here; does this just affirm that interpolating to

a common pressure grid does not introduce additional bias or uncertainty? l. 30-32: Do these extrapolated values actually become part of the profile comparison? If not (as indicated in step 5 on p. 12 l. 10-11), why extrapolate these values at all? If so, profiles with extrapolated values included in the comparison would be problematic: the minimum altitudes for these instruments are so high that they tend to be in the region of the atmosphere where $CH_4$ varies significantly with altitude, and the extrapolation would have large uncertainties. l. 32-33: Is this sentence a way of saying that the averaging kernel equals zero where no measurements exist? Wouldn't this zero out the extrapolation referred to in the preceding sentence?

**Page 12**

l. 16-17: Do you look at the longitudinal variability for each zonal band? If so, does it vary between instruments?

l. 17-20: If my understanding is correct that you apply these filters only for individual points, as opposed to the entire profile, how do you account for heterogeneity in the underlying profile? For example, are all seasons represented at most altitudes ranges? If representation bias is not accounted for, the differences between the underlying measurements might account for some of the features illustrated in Figure 5, e.g. the strong agreement at high vs. low altitudes.

l. 21-22: I find it surprising that measurements within the polar vortex did not impact the results, unless the problematic profiles were discarded through other filters or a priori values were used. Do Holl et al. (2016) apply the same data flags listed in this paper?

l. 28-29: Following on comments made on the manuscript's introduction, an explanation of why zonal biases may exist should be included.

l. 29-31/Figure 5: Reiterating the last general comment, units of ppbv in the left-most panel would remove some of the extra text on the axes associated with the decimals and might be more intuitive for $CH_4$.

l. 31-32: Do the sizes of the bins alter the comparisons? Are these zonal ranges narrow enough in the Northern Hemisphere? i.e. Are the profile differences at 50-60N

comparable to 30-40N?

l. 37: Please give a brief explanation of what this statistic tells us (or justification for using it) as opposed to the general correlation coefficient and/or include a citation.

**Page 13**

l. 18-22: The pressure level at which $CH_4$ decreases is the tropopause height. Unless I am misunderstanding this paragraph, the implication is that the tropopause heights of the instruments are different, which would very likely account for at least some of the profile differences observed. How do the calculated the tropopause heights compare among the various instruments? If they differ, it would indicate that some of the assumptions underlying the pressure interpolation (outlined in the paragraph on p. 11 l. 25-29) might need to be reconsidered. Measurements for which the a priori values have a significant influence could be especially susceptible to tropopause height biases.

l. 24/28: By "below 90 hPa" do you mean less that 90 hPa or at lower altitudes? Similarly, does "Above 100 hPa" mean greater than 100 hPa, or at higher altitudes? Looking at the figure, the reader can deduce the appropriate answer but would benefit from less confusing wording.

l. 24-27: This result seems to me as one of the most important in the manuscript and deserves elaboration. Does the variability have any notable features? Does it depend on sensitivity (s) or a priori influence? Did you find covariance with latitude (i.e. within the 30 degree bins), tropopause height, or season?

l. 33-34/Figure 6: Do zonal differences exist in the unsmoothed data? How do the unsmoothed data fit into the goal mentioned in the introduction for assessing the applicability of Holl et al. (2016) to lower latitudes?

l. 35-36: This sentence is confusing due to the the vague phrasing and verbosity (e.g. "actual differences one would expect"). Please reword.

l. 37: More consistent across instruments? Or across altitudes for each instrument?

**Page 14**

l. 9-11: Why would the differences between ACE-FTS and ESA MIPAS be smaller? Does this shed light on differences between each of these instruments and TANSO?

Section 7.1: Does any of the methodology apply to Section 6 (e.g. criteria for minimizing the dominance of the prior) and vice versa?

l. 21-22/Figure 8: The way this is described, it seems contradictory with the caption on Fig. 8, "The vertical range of partial column integration varies for each pair of co-incident profiles." If you mean that for each coincident measurement pair you match the vertical range of TANSO and each of the other vertical profiles, but that the vertical ranges across all coincident measurements vary, please describe this more explicitly somewhere in this section. Also, if that interpretation is correct, does the vertical range impact the distributions or correlations of the data? (This same question applies to Figure 9.)

l. 34: Why a sensitivity threshold of 0.2? This seems a little low. The minimum of three pressure levels also seems low unless they are contiguous (i.e. don't skip filtered out data in the profile). If the data points do not adjoin each other, did you apply criteria on how far apart the levels can be?

l. 5-7: How different are the results when these 23% are excluded? Do they account for the outliers in Figures 8 and 9?

**Page 15**

l. 10-20: Given that the values are all on a pressure grid, what is the advantage of integrating in altitude/z versus pressure/P? Do you account for water vapour (i.e. use "dry" P/T)?

l. 27: If partial columns with large gaps in the vertical are included (ref. my comment on p. 14 l.34), an uncertainty related with the interpolation should also be propagated through the calculation.

l. 29/Figures 8 and 9: Given the emphasis of the paper on zonal dependencies, please add a colour scale to each plot associated with latitude (bin)?

l. 35-37: What would cause a bias in the intercept? Altitude range? Spectroscopy?

**Page 16**

l. 8/Table 3: Why is the minimum altitude for the NDACC measurements so high? Figure 3 indicates that TANSO is at least somewhat sensitive closer to the surface that 3km.

l. 26: When combining the results, are all data weighed equally, or do you take into account the uncertainties of measurements? Is this the average across all latitudes, or is it a bias that is consistent for all latitudes? Have you also assessed whether altitude-related biases exist in the combined data?

l. 28-29: Did you find a bias in the sub-tropics or mid-latitudes?

l. 31-32: It is not clear what type of comparison was done? Regression? ANCOVA?

**Page 17**

l. 13: The mismatch in vertical extent you point out seems to indicate that these other satellites are not appropriate/useful for validation. If this is not your argument, please rephrase this sentence to make the argument more clear.

l. 15: Have you tried smoothing the TANSO profiles to NDACC to see if the agreement is robust?

l. 18-19: Given these biases, would you recommend "calibrating" the TANSO retrievals?

l. 19-21: Please include how this altitude feature varies (or doesn't) with altitude.

l. 25: It is not clear what "taken over altitude and latitude" means; please reword.

l. 26-27: What improvements are expected in future versions of the retrieval (e.g. priors, spectroscopy)? Based on your results, what would you recommend needs the most/least attention to produce a more accurate data product? Given the limitations of this TANSO product, what applications would it be suited to?

**2 Technical Suggestions**

Comments referring to the addition or removal of punctuation were included where I thought they might improve readability and are thus suggestions rather than corrections, except in cases where the serial comma in a list needs to be added.

**Page 1**
l. 3: Change "$CO_2$ and $CH_4$" to "carbon dioxide ($CO_2$) and methane ($CH_4$)"
l. 15: Change "examine" to "examining"

**Page 2**
l. 3-6: Add VMR (parts per notation) in parentheses after listed percentages.
l. 5: Change "investigated" to "investigate"
l. 7: Unless "over the equator" is specifically what is meant, change to "in the tropics"
l. 29: Remove the comma after "local"
l. 33: Change "made in coincidence" to "coincident"

**Page 3**
l. 30: Add a comma after "surface temperature"
l. 32: Add a comma after "(Maksyutov et al., 2008; Saeki et al., 2013)"

**Page 4**
l. 9: Remove dash after "ACE"
l. 19: Add comma after "Boone et al. (2005)"
l. 24: Remove comma after (Picone et al., 2002)"
l. 25: The use of "and" does not seem correct. Could be replaced with "assimilated into" or "from" depending on the relationship between the met data and the model.
l. 30: Add comma after "profiles"
l. 35: Add comma after "Odin"

**Page 5**
l. 7: Awkward placement of "(inclination of 98)"

l. 9: Add comma after "cloud parameters"
l. 11: Add comma after "2004"
l. 12: Reword end of this sentence, e.g. change ", but" to "with"
l. 24: Add comma after "limb scan"
l. 26: Add comma after "temperature"

**Page 6**
l. 20: Change the commas around "below 25km" to parentheses.

**Page 7**
l. 7: Rephrase "dynamical nature" to a more precise term.
l. 15-16: The use of and phrasing after the semi-colon is awkward and makes the sentence unclear.

**Page 8**
l. 11-12: The parenthetical is awkwardly worded; please revise for clarity.
l.12: Change "differences is also" to "differences are also"
l. 13: Please reword "When examining dates with several measurements" to make more clear.
l. 16-17: The grammatical structure of this sentence is difficult to follow. Please reword.

**Page 9**
l. 32: Add comma after "differ"

**Page 10**
l. 10: Change comma after "kernel" to semi-colon.
l. 13: Add comma after "role"
l. 20: Add comma after "altitudes"
l. 22: Change "70km. This is shown in Fig. 3e." to "70km (Fig. 3e)."
l. 22: Add comma after "development"
l. 27-28: Please reword. The structure of this sentence is difficult to follow, e.g. the verb ("are shown") appears twice and each has its own modifier.

**Page 11**

l. 22-23: Remove parentheses, and add a period or semi-colon after "retrieval"

l. 24: Please state which retrieval you're referring to (seems like the higher-resolution profile, but not self-evident).

l. 27: Remove comma after "equilibrium"

**Page 12**

l. 21-22: Change "looking for" to "filtering" and remove ", and then filtering these events," to make the sentence more clear.

**Page 13**

l. 20: I think a verb is missing after "VMR decrease" (e.g. "occurs").

l. 33: Add a comma after "zonally"

**Page 16**

l. 29: Change "or" to "and" ("0.014 ppmv or 0.020 ppmv"), and "Pole" should be plural.

**Page 17**

l. 10: Remove commas after "sensitivity" and "product"

l. 12: Remove comma after "altitudes"

l. 12-13: Please reword "and that there is a limitation on the useful upper altitude of its data product of below 15 or 20km" to follow the clarity and structure of the beginning of the sentence.

l. 14: Add "upper" before "troposphere" (Without the addition, this sentence is misleading.)

l. 20: Phrase starting with "and in a consistent manner" needs rewording for clarity.

**Figures**

Figures 2 and 10: The legend has two icons for every instrument, which adds extra visual clutter.

Figure 3: Several line colours do not appear in the legends of a-d. Instead of using a legend, you might consider labelling each line with the pressure using the same colour

for the text.

Figure 7: A heat map or similarly sequential colour scheme could be more helpful for this type of plot.

Figure 8: The "R" is missing on the $R^2$ line of each sub-figure, and it looks like some other letters and numbers might also be missing.

Figure 10: The degree symbol is missing between parentheses on the x-axis label. Also, please add additional tick marks on the x-axis. You might consider including light gray grid lines behind the data.

---

## Referee Comment (RC2) · Anonymous Referee #1 · 3 Jun 2017

General comments: This paper describes a comparison of CH4 profiles retrieved from the GOSAT TANSO-FTS TIR with measurements by ACE-FTS, ESA MIPAS, IMK-IAA MIPAS, and NDACC. Although this manuscript presents results that would be of interest to readers of AMT, I found some of the authors' explanations difficult to follow. Therefore, some revisions are needed before it can be accepted for publication.

[1] p1, line14-15: "with and without smoothing" p9, line13: "To reduce biases caused by over-counting, when comparing TANSO-FTS to MIPAS, and by smoothing, when comparing TANSO-FTS to ACE-FTS,..." What is "smoothing" in this study? Please

add a detailed description in Abstract and text to help the readers. Additionally, the authors should explain why they show correlation results based on both smoothed CH4 profiles (Fig. 8) and unsmoothed CH4 profiles (Fig. 9). What can we learn from this comparison?

[2] p7, line13: "internal variability for each instrument" Due to insufficient description, I don't understand the meaning of "internal variability" in Sect. 3 and Fig. 1. Green lines (TANSO-FTS) in Fig. 1 show the difference between the GOSAT TANSO-FTS CH4 retrievals and the a priori profiles. On the other hand, blue lines (MIPAS) are the difference between IMK-IAA MIPAS and ESA MIPAS. I don't understand how were the internal variabilities of ACE-FTS (p7, line25-33) and NDACC (p8, line9-15) evaluated. Does "the variability of NDACC data" mean the difference between NDACC CH4 profile and TANSO-FTS CH4 profile? In addition, can the authors explain the reason why they were compared in the same figure despite a different definition?

[3] p8, line20-27: "coincidence criteria" There is a lack of explanation why the coincident criteria were set as "within 12 hours and within 500km" for ACE-FTS and NDACC and set as "within 3 hours and within 300km" for the MIPAS data. For example, did the authors examine latitudinal and longitudinal dependence of TANSO-FTS data within 500km or 300km? I would show the spatial variations of TANSO-FTS CH4 in the co-location circle at a particular height (the upper or middle troposphere). In addition, can the authors discuss the validity of their method by comparing the coincidences (e.g., statistics for match-upped data) in present study to those in the previous validation papers on the GOSAT data.

[4] p16, line31-34 "We also compared the differences shown in Fig. 10 to TANSO-FTS retrieval parameters: land or sea mask, sunglint flag, incident angle, both along the scan path and GOSAT track path, and observation mode (see Kuze et al., 2009). We found no biases in our coincident TANSO-FTS dataset related to any of these parameters, or whether the observation was made during night or day." Can the authors show the features of the GOSAT TANSO-FTS biases related to land or sea mask and

the other parameters in the previous section (or in Appendix)? It is not appropriate to discuss these important points without showing here.

Other minor revisions: [1] p4, line32: "the Halogen Occultation Experiment" —> "the Halogen Occultation Experiment (HALOE)" [2] p7, line38: "the IMK-IAA data has" —> "the IMK-IAA data have" [3] p12, line23: "have a much smaller affect on" —> "have a much smaller effect on"? [4] p15, line34: The Pearson correlation coefficient R2 of NDACC (0.9929) is different from that shown in Fig. 8. [5] p19, line13: Please update information on Bader et al., 2016, ACPD. [6] p19, line10: in reference list of Côté et al. (1998), "formulations" —> "formulation" [7] p20, line19: Please update information on Errera et al., 2016, AMTD. [8] p21, line33: in reference list of Picone et al. (2002), "1486" —> "1468" [9] p21, line11: in reference list of Raspollini et al. (2014), "Annal. Geophys.," —> "Ann. Geophys.," [10] p28: a legend of Fig. 2, "NDDAC" —> "NDACC" [11] p34: In Figs. 8 and 9, "x" of "y = mx+b" is not printed. In addition, "R" of "R2" is not printed. [12] p35: In Fig. 10, the unit of "Latitude" is not printed.

---

## Author Response (AR1)

**Authors' response to: "Comparison of the GOSAT TANSO-FTS TIR $CH_4$ volume mixing ratio vertical profiles with those measured by ACE-FTS, ESA MIPAS, IMK-IAA MIPAS, and 16 NDACC stations"**

Kevin S. Olsen[1], Kimberly Strong[1], Kaley A. Walker[1,2], Chris D. Boone[2], Piera Raspollini[3], Johannes Plieninger[4], Whitney Bader[1,5], Stephanie Conway[1], Michel Grutter[6], James W. Hannigan[7], Frank Hase[4], Nicholas Jones[8], Martine de Mazière[9], Justus Notholt[10], Matthias Schneider[4], Dan Smale[11], Ralf Sussmann[4], and Naoko Saitoh[12]

[1]Department of Physics, University of Toronto, Toronto, Ontario, Canada
[2]Department of Chemistry, University of Waterloo, Waterloo, Ontario, Canada
[3]Istituto di Fisica Applicata "N. Carrara" (IFAC) del Consiglio Nazionale delle Ricerche (CNR), Florence, Italy
[4]Institut für Meteorologie und Klimaforschung, Karlsruhe Institute of Technology, Karlsruhe, Germany
[5]Institute of Astrophysics and Geophysics, University of Liège, Liège, Belgium
[6]Centro de Ciencias de la Atmósfera, Universidad Nacional Autónoma de México, Mexico City, Mexico
[7]Atmospheric Chemistry Division, National Center for Atmospheric Research, Boulder, CO, USA
[8]Centre for Atmospheric Chemistry, University of Wollongong, Wollongong, Australia
[9]Belgisch Instituut voor Ruimte-Aëronomie–Institut d'Aéronomie Spatiale de Belgique (IASB-BIRA), Brussels, Belgium
[10]Institute for Environmental Physics, University of Bremen, Bremen, Germany
[11]National Institute of Water and Atmospheric Research Ltd (NIWA), Lauder, New Zealand
[12]Center for Environmental Remote Sensing, Chiba University, Chiba, Japan

*Correspondence to:* K. S. Olsen (ksolsen@atmosp.physics.utoronto.ca)

**1 Anonymous Referee #1**

**1.1 General comments**

This paper describes a comparison of $CH_4$ profiles retrieved from the GOSAT TANSO-FTS TIR with measurements by ACE-FTS, ESA MIPAS, IMK-IAA MIPAS, and NDACC. Although this manuscript presents results that would be of interest to readers of AMT, I found some of the authors' explanations difficult to follow. Therefore, some revisions are needed before it can be accepted for publication.

We thank the reviewer for their careful review. We will address each specific comment and try to clarify our manuscript as well as possible. Our aim was to provide a clear and concise description of our methods and focus on our results, which we also believe are of interest to the community.

**1.2 Specific comments**

1. p1, line14-15: "with and without smoothing" p9, line13: "To reduce biases caused by over-counting, when comparing TANSO-FTS to MIPAS, and by smoothing, when comparing TANSO-FTS to ACE-FTS,. . ." What is "smoothing" in this study? Please add a detailed description in Abstract and text to help the readers. Additionally, the authors should explain why they show correlation results based on both smoothed $CH_4$ profiles (Fig. 8) and unsmoothed $CH_4$ profiles (Fig. 9). What can we learn from this comparison?

Smoothing is a general term used to describe an operation that reduces the magnitude or frequency of fine-scale structure in a signal. When comparing two atmospheric remote sensing instruments with different vertical resolutions, the instrument with finer vertical resolution will have more fine-scale structure in its retrieved profiles. Likewise, an instruments whose retrieval has less of a dependence on the a priori may also have more fine-scale structure in its retrieved profiles. The instrument with lower vertical resolution or more dependence on the a priori will have retrieved vertical profiles that look "smoother." In order to compare the results of two instruments that are intrinsically different, we apply smoothing to that with finer resolution in order to account for these instrumental differences. Our objective is not to compare the results from two different instruments, but to ask, if one of our instruments had the same vertical resolution, information content, and dependence on the TANSO-FTS a priori, would the retrievals for that instrument agree with those from TANSO-FTS. The process, as described by Rodgers and Connor (2003), is standard practice when validating remote sensing vertical profiles of trace gas VMRs. The purpose of smoothing and the method used are described in Sect. 6.1. At the first mention of "smooth" in the abstract, the sentence has been extended to include a brief description of the purpose of smoothing. In the introduction, a few sentences were added to describe the need for smoothing and refer to Sect 6.1 for the method.

The reason we present results with and without smoothing is that a data user may not apply smoothing to the ACE-FTS, NDACC, or MIPAS results. The objective of validation is not necessarily to measure the magnitude of the differences between the two instruments' retrievals, but to do so in the context of the sensitivity and information content of the instrument being validated. These results may be of interest to data users, so they have been included.

2. p7, line13: "internal variability for each instrument" Due to insufficient description, I don't understand the meaning of "internal variability" in Sect. 3 and Fig. 1. Green lines (TANSO-FTS) in Fig. 1 show the difference between the GOSAT TANSO-FTS $CH_4$ retrievals and the a priori profiles. On the other hand, blue lines (MIPAS) are the difference between IMK-IAA MIPAS and ESA MIPAS. I don't understand how were the internal variabilities of ACE-FTS (p7, line25-33) and NDACC (p8, line9-15) evaluated. Does "the variability of NDACC data" mean the difference between NDACC $CH_4$ profile and TANSO-FTS $CH_4$ profile? In addition, can the authors explain the reason why they were compared in the same figure despite a different definition?

Several changes have been made to the internal variability section to address your concerns. This study is not central to the paper, but we feel that it is important to provide context for the main results. The internal variability is the difference between measurements within each instruments data set, loosely defined (too much so, we agree). Due to the different measurement techniques (especially the data acquisition rate of ACE-FTS compared to TANSO-FTS and MIPAS), a single method to estimate this variability was not used. Furthermore, there were other calculations of interest to us, for instance, the IMK and ESA data products for MIPAS allow us to directly compare different retrievals made from the same observations, something we cannot with ACE-FTS or NDACC. The measurements compared in this study are made at different times and locations, sampling different air-masses, and are subject to noise, and considering the internal variability of each instrument addresses the magnitude of the effects caused by these differences. Several changes were made to make the purpose of this study clear and we have tried to eliminate our usage of the term "internal variability" since we are presenting different measurements. Changes were also made to the caption of Fig. 1 to reflect this.

The reason each variability profile is placed on the same figure is because they are to be qualitatively compared and we see no reason to unnecessarily create extra figures and paper length.

3.      p8, line20-27: "coincidence criteria" There is a lack of explanation why the coincident criteria were set as "within 12 hours and within 500km" for ACE-FTS and NDACC and set as "within 3 hours and within 300km" for the MIPAS data. For example, did the authors examine latitudinal and longitudinal dependence of TANSO-FTS data within 500km or 300km? I would show the spatial variations of TANSO-FTS $CH_4$ in the colocation circle at a particular height (the upper or middle troposphere). In addition, can the authors discuss the validity of their method by comparing the coincidences (e.g., statistics for match-upped data) in present study to those in the previous validation papers on the GOSAT data.

The coincidence criteria were used to try to optimize the number of coincidences with TANSO-FTS, increasing the small number of NDACC and ACE-FTS coincidences, and reducing the large number of MIPAS coincidences. Because MIPAS makes measurements much more frequently, we have the freedom to demand much tighter coincidences in space and time. At the beginning of Sect. 4, where "coincident criteria" is used, this is made explicitly clear with the statements: "Due the coverage and data collection rates of each instrument, different coincidence criteria were used. In the case of ACE-FTS, which only records two occultations per orbit, and NDACC stations, which are stationary, the objective of the coincidence criteria was to maximize the number of measurements used. Conversely, in the case of MIPAS, which makes frequent observations, the objective was to reduce the number of potential coincident measurements." Furthermore, we point out: "...for MIPAS–TANSO-FTS coincidences within 12 hours and 500km, we find approximately 180,000 coincidences per month."

The TANSO-FTS data are collected with a high frequency in a sweeping pattern along the satellite ground track. The high inclination of $98°$ provides a near-polar orbit. The result is high-density, near-global coverage, with more observations near the poles because the satellite ground tracks are more tightly spaced at higher latitudes. Reducing the spatial dependence of the coincidence criteria will have a different effect on each satellite. The impact will be larger on ACE-

FTS because its measurements over the tropics are very sparse, but for ACE-FTS we used the wider criteria. A way to avoid this difference in sample sizes between the tropics and poles is to use a degrees-latitude criteria. The result is that over the poles we are comparing measurements that are very close together, while those over the tropics may be separated by hundreds to thousands of kms, which would have a larger negative impact on our study.

The criteria used match other validation studies that use ACE-FTS, NDACC, and MIPAS data, and also the validation studies of $CH_4$ for each instrument. These are, however, far too numerous to list here. Coincidence criteria for primary $CH_4$ validation studies have been added to the text.

The reviewer asks whether we can compare coincidence statistics to previous validation papers on the GOSAT data. Only one previous validation paper on TANSO-FTS TIR data is in press: Saitoh et al. (2015) compares the $CO_2$ data product to aircraft measurements. They use very tight criteria (100 km and 2 hr) and consider only 140 coincident profiles. However, this study is not comparable with ours since the aircraft flights are in-situ measurements, rather than remote sensing, so a tighter coincident criteria is needed. The TANSO-FTS SWIR $XCH_4$ data product was validated by Inoue et al. (2014) that included the ACE-FTS instrument. However, they use climatological data, not coincidences.

4.  p16, line31-34 "We also compared the differences shown in Fig. 10 to TANSO-FTS retrieval parameters: land or sea mask, sunglint flag, incident angle, both along the scan path and GOSAT track path, and observation mode (see Kuze et al., 2009). We found no biases in our coincident TANSO-FTS dataset related to any of these parameters, or whether the observation was made during night or day." Can the authors show the features of the GOSAT TANSO-FTS biases related to land or sea mask and the other parameters in the previous section (or in Appendix)? It is not appropriate to discuss these important points without showing here.

The reviewer asks us to show the features of the GOSAT TANSO-FTS biases related to land or sea mask." However, there were no biases to show. We understand the reviewers comment that it may be inappropriate to have mentioned investigating these quantities without explicitly showing the results. Our objective is to show due diligence on our part. We are not making strong statements about the effects of any of these quantities on the TANSO-FTS data, and that is not the purpose of this paper. We feel that it may be detrimental to our manuscript to include several more figures and significantly increase its length, while not significantly adding to the conclusions of our work. An appendix may be created showing the relationships of our results to the ancillary data in the GOSAT data files (eight additional figures) at the discretion of the editor.

**1.3 Minor revisions**

1.  p4, line32: "the Halogen Occultation Experiment" —> "the Halogen Occultation Experiment (HALOE)"

change made, thank you

2.  p7, line38: "the IMK-IAA data has" —> "the IMK-IAA data have"

changed "has" to "have" for the plural of "data"

3.  p12, line23: "have a much smaller affect on" —> "have a much smaller effect on"?

this is certainly a case where the noun "effect" is correct

4.  p15, line34: The Pearson correlation coefficient R2 of NDACC (0.9929) is different from that shown in Fig. 8.

thank you, this is a typo. The figure is correct, $R^2$ is computed during the generation of the figure. We also changed the order of the list to match the order of the panels of the figure

5.  p19, line13: Please update information on Bader et al., 2016, ACPD.

updated the reference.

6.  p19, line10: in reference list of Côté et al. (1998), "formulations" —> "formulation"

corrected the typo in the title

7.  p20, line19: Please update information on Errera et al., 2016, AMTD.

updated the reference

8.  p21, line33: in reference list of Picone et al. (2002), "1486" $\rightarrow$ "1468"

changed the page number from 1486 to 1468

9.  p21, line11: in reference list of Raspollini et al. (2014), "Annal. Geophys.," $\rightarrow$ "Ann. Geophys.,"

corrected the journal abbreviation

10. p28: a legend of Fig. 2, "NDDAC" $\rightarrow$ "NDACC"

corrected the typo. Similar to the next two items, axis labels indicating cardinal direction are missing from manuscript version downloaded from the AMTD website

11. p34: In Figs. 8 and 9, "x" of "y = mx+b" is not printed. In addition, "R" of "R2" is not printed.

during the technical review of this manuscript, this problem was mentioned. However, this issue did not exist on our copies of the submitted manuscript. We have downloaded copies from the AMTD website, and, sure enough, characters have been stripped from the figure. This is not an issue we can change, but will have a special note to the editors upon re-submission and ensure that the correct figure ends up in the compiled document. Thank you for pointing this out

12. p35: In Fig. 10, the unit of "Latitude" is not printed.

similar as for Fig. 9, the degree symbol appears in our submitted manuscript and figure files. We will discuss with the editors and ensure this symbol appears in a future version

**2 Anonymous Referee #2**

**2.1 General comments**

Olsen et al. (2017) compares GOSAT TANSO-FTS profile retrievals and partial columns of methane (CH4) to coincident data from ACE-FTS, ESA MIPAS, IMK-IAA MIPAS, and NDACC, focusing on the upper troposphere/lower stratosphere (UTLS) over 2009-2013. This work expands an earlier TANSO inter-comparison in the Arctic to include global measurements from additional satellite and ground-based remote sensing instruments, aiming to identify possible zonal variability in the profile retrievals. Given the importance of understanding global CH4 and the uncertainty in its recent trends, precise and accurate measurements with global coverage are very much needed. As GOSAT is one of the primary satellites in monitoring atmospheric CH4 today, the as- sessment of the spatial biases of TANSO measurements would give confidence to the scientific community's use of an important data set. The approach described in the paper is comprehensive, and the methodology is thought-through. However, the exposition of the significance, conclusions, and limitations of this work require more development. The paper would benefit from a rebalancing of its structure, with the most critical changes being: (1) augmenting the discussion of previous and/or similar validation efforts of TANSO CH4, (2) paring down of the instrument background sections to focus on details directly relevant to their results, and (3) discussing the reasons for and implications of the differences highlighted in the comparison. Therefore, I would recommend publication of this manuscript after these points are addressed.

We thank the reviewer for taking time to make such a thorough review of this manuscript. Their comments are about the general structure and language of the paper, and cover specific technical and grammatical points. We have worked very hard to address each comment with care and believe that the outcome is a more streamlined paper that better addresses the scientific merit of our study without distracting the reader with overly technical discussions. The three primary points made above are discussed in more detail below by the reviewer, asking pointed questions and offering examples. After careful consideration, we agree with the reviewer and have made several changes and additions to the manuscript,

**2.2 Scientific evaluation general comments**

1.  This paper would benefit from a more rigorous analysis (or if already done, a more comprehensive description) of the causes of the differences noted in the results. The authors are thorough in considering different parameters, but the text lacks a synthesis of how these difference relate to the results of the comparisons. For example, what component of the differences between TANSO and the other instruments can be explained by spectroscopy verses a priori profiles used? This can be addressed for the specific parameters already mentioned in the text; e.g. many of the retrievals incorporate the same linelists–do those profiles show better agreement to each other? The implications of these differences for the scientific community's use of the TANSO product based on these differences would also strengthen the paper.

Some of the primary comments from reviewer #1 are relevant to Sect. 3 where we look at the variability in each dataset. A discussion of these results in the context of our inter-instrumental comparison is lacking. We agree that the text lacks a synthesis and have made significant changes to Sect. 3 and added text to our discussion in Sect. 8. We think this reasonably addresses the reviewers comments and strengthens our manuscript.

Some of the specific questions raised in this reviewer's comment are, we feel, beyond the scope of this research topic and manuscript. Such aspects as the TANSO-FTS retrieval's dependence on the a priori or line list can be insinuated from our work, but not properly quantified. We have made a better effort to comment on our results in Sect. 8, but, since we cannot vary these parameters, redo the retrievals, and test the effects of these parameters, thoroughly addressing these questions is best left to a future, specific manuscript about the TANSO-FTS TIR $CH_4$ vertical profile retrievals.

The impact of differences in spectroscopy and retrieval methods on our results is difficult to quantify, but these factors certainly make significant contributions. For example, the retrievals for the instruments we considered use different versions of the HITRAN line list. There are significant and much discussed differences between $CH_4$ line lists in each version and $CH_4$ spectroscopy may be a limiting factor in $CH_4$ retrieval accuracy. The next version of the TANSO-FTS TIR $CH_4$ data product may use a different version of HITRAN and the TANSO-FTS team will quantify the impact of the line lists.

We quantified the difference between the IMK-IAA and ESA MIPAS retrievals by comparing pairs of profiles retrieved from the same MIPAS observations. This shows the impact of the retrieval algorithm and spectral line lists. The discussion of this in the manuscript has been improved to highlight this. The largest effect on our study, however, is due to physics and instrument design. TANSO-FTS is a much lower resolution instrument than MIPAS, ACE-FTS and those used by NDACC. Viewing geometry is what led to the use of a retrieval algorithm that incorporates the a priori into its solution, and also results in smaller signal strength than the other instruments. Our discussion has been extended to more thoroughly discuss our findings in Sect. 3, and to address the differences in instrument capability and retrieval algorithms.

2.       At several points throughout the paper, principally in Sections 2-5, the authors include technical details about the satellites/instruments that do not seem pertinent to the study presented (e.g. the inclination angles). The extraneous information was distracting and diluted the narrative of the instrument comparison. I recommend including only those details that the reader should know to understand and evaluate the results and conclusions of this comparison, especially if those details are published elsewhere. For necessary details where the connection to the present study is not clear, explicitly listing the relevance and/or implications would be useful. (e.g. The authors list measurement windows and spectral resolution but need to comment on how these inter-instrument differences might relate to the results.) The authors might also consider moving some of these details that are useful but not central to the paper into an appendix or supplement.

While we believe these details are necessary for the reader, we understand the drawbacks pointed out by the reviewer, especially thanks to several specific examples. We removed the ranges of the non-TIR TANSO-FTS channels, the details

about MAESTRO on SCISAT, details about TANSO CAI, and the inclination of MIPAS. The inclination of SCISAT is important, since it affects the location of solar occultation measurements which leads to far more coincidences at high latitudes. The inclination has been moved to the coincidence discussion, Sect. 4, and follows a statement about the distribution of the ACE-FTS measurements. In the discussion, Sect. 8, we have addressed the different spectral ranges and resolutions, and their possible impact on our study.

3.        Given that methane is generally provided in units of ppb, do the authors have a specific reason for using ppm? If not, I suggest changing the references and figures to ppb. Figure 5 in particular would be more clear without extraneous zeros and decimal places.

When discussing methane on Earth, the abundance in the troposphere is around 1.5–1.7 ppmv, and the literature commonly uses ppmv when discussing $CH_4$. Our work is consistent with other, related literature sources, such as the $CH_4$ validation studies for MIPAS and ACE-FTS (Payan et al., 2009; Pleininger et al., 2015; de Maziere et al., 2008) and the preceding work by Holl et al. (2015). That being said, we do agree with the reviewer about Fig. 5 and have changed its $x$-axis from ppmv to ppbv. We have left Figs. 1 and 6, since we would be adding zeros to the $x$-axes of those graphs. Furthermore, we agree that sticking to ppmv when discussing $CH_4$ *differences* (e.g., Sect. 8) is cumbersome, and we have made several changes from ppmv to ppbv in the text.

**2.3   Specific comments**

1.        p. 2 l. 11-13: The redundancy in the list of GOSAT objectives can be pared down. In addition, these objectives should be related, at least in part, to the objectives of this research. i.e. How does this paper contribute to the objectives of the satellites and the scientific community? This question is briefly touched upon on l. 31, but needs to be developed.

We have broken the short paragraph at line 29 up, placing the argument mentioned at line 31 with the GOSAT objectives at lines 10-13.

2.        p. 2 l. 24-26: Given the focus on zonal dependence, providing the latitude of Eureka in the text would be useful. Also, please include citations for PEARL and NDACC.

We provided the altitude of Eureka and added two citations.

3.        p. 2 l. 29-31: As the main objective (and contribution) of this paper is to expand TANSO validation globally, more consideration of the issues of spatial coverage is needed, including a literature review of zonal biases (possibly does not exist for this particular TANSO product, but if that is the case this should be stated) and a description of mechanisms that make the Arctic non-representative (e.g. polar vortex changing vertical profiles of trace gases and reducing the accuracy of a priori information).

This paragraph was removed as part of the correction for comment 1. Such a study for the TANSO-FTS data was performed by (Holl et al., 2016) and his results are discussed in Sect. 6.1.

4.      p. 3 l. 5/Table 1: Please add year ranges to Table 1. It is not clear when mentioning the 2009-2013 time frame in the abstract whether all of the instrument measure consistently over that time period.

This is not so easy to address, all the instruments have downtime, and those operated at high latitudes are run in seasonal campaigns. Only two of the instruments came online during our study, Altzomoni and La Reunion Maido, and those dates have been added as footnotes to Table 1.

5.      p. 3 l. 7-14: If this is an outline, put section numbers after each sentence. If this is an overview, this paragraph might fit better in the methods section rather than the introduction.

Appropriate section numbers are given on lines 6-7 and 15-19.

6.      p. 3 l. 27: What type of coverage? Spatial? Spectral?

This is spatial coverage, added the qualifier to the text.

7.      p. 3 l. 28: Is this paper the first time methodology for CO2 is applied to CH4? If so, the retrieval (or the at least the aspects that differentiate it from other TANSO retrievals) should be described more.

We added more details about the algorithm used, which is more in-line with our presentation of the MIPAS and NDACC data sets.

8.      p. 3 l. 33: Do the 2011 updates have a reference (e.g. on the HITRAN website or used in a validation paper)? If so, please include a citation.

There is not a citation for this. The HITRAN 2008 paper refers users to the HITRAN website.

9.      p. 4 l. 4-6: This sentence, with the important conclusions of the referenced paper added, should be moved to the Introduction, at the end of the second paragraph.

This is a summary of what is already in the introduction. Page 2, lines 22–28 discuss this in greater detail and provide the actual, quantitative conclusions.

10.      p. 4 l. 9-11: If information about MAESTRO is relevant, include a reference for the instrument; otherwise this sentence can be removed.

Technical details about MAESTRO have been removed.

11.      p. 4 l. 16: Is 5km the lowest altitude for ACE-FTS measurements filtered using the recommended flags (e.g. not a priori values)? Listing the lowest altitude of the data used in this paper would be more relevant, particularly for the discussion on vertical range in subsequent sections.

5 km is approximate, some occultations may extend lower. As can be seen in Figs. 5 and 6, our data extend to 6 km.

12.      p. 4 l. 33 and 36: Do the percentages listed apply to all trace gases or just methane? Please make this more clear.

Added specific references to $CH_4$. The validation paper de Maziere et al., (2008) focuses only on $CH_4$.

13.  p. 5 l. 4-5: Because this is the data version used, the results from Waymark et al. (2013) should be summarized, as is done with the above papers.

Waymark et al., (2013) contains a table summarizing the results fro each gas. The result for $CH_4$ was added to the manuscript.

14.  p. 5 l. 25: By "Initial guesses" do you mean a priori profiles? Please clarify.

The ESA MIPAS team makes a distinction between "initial guess" and a priori. The initial guess is a generic profile used to begin the least squares fitting iterations. The term a priori means that we have prior knowledge of the state of the atmosphere and that the initial profile has a level of accuracy that the "initial guess" may not.

15.  p. 6 p. 5l. 6-7: What are the reasons for the outliers/discontinuities? Would these impacts the results of this study?

Outliers would hopefully have little effect on this study due the large sample size we use. We rely on the ESA MIPAS data quality flags to remove spurious results.

16.  p. 6 l. 30: Using "some information" is vague and does not tell the reader the relevance of the included information.

We agree, "some information" is vague. That may have been written before we decided on what information to include in Table 1. The specific columns are now in place of the vague terminology.

17.  p. 6 l. 32-35: Is there a reference for an inter-comparison of these instruments? How are inter-site differences due to different instrumentation accounted for in this study?

There is not a publication detailing exactly what the reviewer asks yet, but the reader is invited to see Table 1 for publications describing each site.

18.  p. 6 l. 36: Are those references for the most recent versions of the retrieval software? The spectroscopy has presumably changed since 1995/2004.

These references are not for the most recent versions of the software, but are the most recent publications about the software.

19.  p. 6 l. 37: Does "harmonized" mean consistent between sites? Please use a more clear term.

Yes, harmonized means to "make consistent or compatible," so in our context we mean that the site operators have attempted to make the retrievals consistent between sites. This is the terminology used by the NDACC InfraRed Working Group.

20.        p. 7 l. 10: Given the differences delineated in this section, have you done a covariance analysis or sensitivity test to assess whether the results of the comparison depend on the retrieval software used, instrument, or any other difference across NDACC sites?

Please see line 11, the start of Sect. 3. This comment is addressed as part of the reviewers Scientific Evaluation.

21.        p. 7 l. 20: Are these measurements representative of year, season, location, tropopause height, etc.?

These data were randomly selected from our sample to avoid diurnal, seasonal, or zonal effects. This information has been added to the text.

22.        p. 7 l. 25: Please define "sunset/sunrise measurement" in this context.

Noted that sunset and sunrise refer to the occultation direction. In solar occultation geometry, the satellite instrument makes a series of observations by directly viewing the sun while the limb of atmosphere intersects the line of sight. In each orbit there are two occultation opportunities: when the satellite enters the shadow of the Earth, and when the satellite exits the shadow of the Earth. The instrument observes sunsets and sunrises, respectively.

23.        p. 8 l. 5-6: Please summarize the results that the bias is consistent with.

This is in Sect. 2.3.2 where Laeng et al. (2015) is summarized.

24.        p. 8 l. 24: Why was reducing coincident measurements an objective, if you could average them and thereby reduce potential bias (c.f. Kulawik et al. 2016)? Was this a data processing issue from the large number of coincidences? If the coincidence criteria varies by instrument, some sort of bootstrap or sensitivity test with a subset of data should be run to see if the VMRs are different with the more lax coincident criteria.

For consistency, we wanted to have sample sizes for the MIPAS comparison to be of the same order of magnitude as those for NDACC and ACE-FTS.

25.        p. 9 l. 8-9: Including references for the approach mentioned would be useful.

Added a reference to Holl et al. (2016) for example.

26.        p. 9 l. 19: Please define a z score and/or include a reference.

$z$-score is a standard parameter in statistics found in most undergraduate textbooks. It is perhaps more appropriately referred to as the "standard score," and it has been changed.

27.        p. 9 l. 26-27/Figure 2: Does using only the first 200 observations of the year capture any time-varying spatial coverage of the satellite data? Would it be more appropriate to use the first 20 observations of each month of 2012, for instance?

The orbit actually repeats its flyover-locations every couple of weeks. Using the first twenty occultations from each month would introduce a bias. However, the figure is purely qualitative. It shows the NDACC sites, that ACE-FTS favours high latitudes, and that there are no ESA MIPAS coincidences over the tropics.

28.        p. 10 l. 5-6/Figure 3: Please provide a more clear description of what the pressure levels in the legends correspond to (i.e. the pressure widths of the averaging kernel rows vs. the pressure on the y-axis), as relates to the findings of the paper. (You could perhaps include the simple averaging kernel equation if useful, but if the text becomes too detailed I would suggest moving this description to an appendix/supplement.) Also, if these pressure levels are meant to be compared across instruments, using a single colour scale for panels a-d would be helpful, e.g. binned into ranges or following a colour gradient.

Since pressure is a retrieved parameter for some of the instruments, each retrieval is done on a unique pressure grid. To compute means of the averaging kernels, each kernel has to have the same vector length corresponding to a common pressure grid. Since Fig. 3 is illustrative, the pressure grid for each panel is not important, but that the means are taken at the same pressure level is. The pressure levels of the averaging kernels are given in each data product. A sentence was added to clarify this. The colours indicating each instrument are consistent throughout the manuscript.

29.        p. 10 l. 8-12: Please add the implications for these comparisons. Related to the previous comment, the remarks on "full-width at the half-maximum values" would be more understandable by rewording the phrase "values when considering the location of the appropriate pressure level" and adding a more clear description of the averaging kernel widths (p. 9 l. 33-34).

The implication of these findings is given on line 8, that TANSO-FTS is the instrument with coarser vertical resolution, and on line 25 after further description of the figure, that TANSO-FTS retrievals are highly dependent on their a priori. We've re-organized this paragraph to be more clear, introduced FWHM notation, and moved the definition of vertical resolution to the discussion of kernel widths.

30.        p. 10 l. 25-26: How do you determine the influence of this dependence on the results? (This paragraph might need to be moved closer to the discussion of TANSO priors later in the manuscript.)

This paragraph is a description of Fig. 3 and follows the preceding discussion of MIPAS and ACE-FTS sensitivity naturally. The implication is explored further later in the manuscript, especially in the discussion, Sect. 8, which was greatly expanded to address comments from reviewer #1.

31.        p. 10 l. 32: What is the implication of the flat trends over mid-latitudes and tropics, as relates to the objective of this paper to determine zonal dependence of the retrievals?

The trends are related to seasons, their southern hemisphere reversal is actually the phase difference of the seasons. This has been re-written to reflect that.

32.        p. 11 l. 28: Please include a reference for the claim that the NDACC a priori/ measured pressure profiles are accurate. (This might fit better in Section 2.)

Added a reference to the NDACC retrievals paper (Sepúulveda et al., 2014), which quantifies the contributions of sources of error, including NCEP a priori.

33.      p. 11 l. 27-29: I'm not sure I follow the logic here; does this just affirm that interpolating to a common pressure grid does not introduce additional bias or uncertainty?

What we are defending here is not the choice of pressure grid, but the choice of doing our analysis in terms of pressure levels instead of altitude. We argue that pressure is more well known than altitude. We have added a statement specifying this.

34.      p. 11 l. 30-32: Do these extrapolated values actually become part of the profile comparison? If not (as indicated in step 5 on p. 12 l. 10-11), why extrapolate these values at all? If so, profiles with extrapolated values included in the comparison would be problematic: the minimum altitudes for these instruments are so high that they tend to be in the region of the atmosphere where CH4 varies significantly with altitude, and the extrapolation would have large uncertainties.

A common vertical grid is vital since we are finding the means of profiles. Extrapolation is needed because the vector lengths need to match those of TANSO-FTS averaging kernels. Extrapolated pressure levels are not used. A statement was added that explains the need for extrapolation and refers to Eq. 1.

35.      p. 11 l. 32-33: Is this sentence a way of saying that the averaging kernel equals zero where no measurements exist? Wouldn't this zero out the extrapolation referred to in the preceding sentence?

In the non-overlapping region, we have set values of $\hat{x}$ equal to $x_a$. When $\mathbf{A}$ is multiplied by $(\hat{x} - x_a)$, $(\hat{x} - x_a)$ nullifies the contributions from the averaging kernel matrix at pressure levels where $(\hat{x} - x_a)$ is zero.

36.      p. 12 l. 16-17: Do you look at the longitudinal variability for each zonal band? If so, does it vary between instruments?

The longitudes of coincidences are fairly uniformly distributed in each zone for ACE-FTS and MIPAS, but are not for NDACC.

37.      p. 12 l. 17-20: If my understanding is correct that you apply these filters only for individual points, as opposed to the entire profile, how do you account for heterogeneity in the underlying profile? For example, are all seasons represented at most altitudes ranges? If representation bias is not accounted for, the differences between the underlying measurements might account for some of the features illustrated in Figure 5, e.g. the strong agreement at high vs. low altitudes.

We use the data products as instructed by the product documentation, in some cases, specific points, usually at altitude extrema, are discarded, in others, the entire profile is discarded (e.g., when we say the ACE-FTS *quality_flag* cannot be equal to 4 at any altitude). Seasonal effects are smaller than zonal effects, the strongest being the increased cloud and humidity over the tropics, which strongly reduces the number of overlapping coincidences from MIPAS. Seasonal effects are shown in Fig. 4, and the strongest effect is for IMK MIPAS. The decrease in IMK MIPAS DOFs in Arctic winter corresponds to fewer measurements in the observation (ESA MIPAS, TANSO-FTS, and ACE-FTS do not have such a

seasonal dependence). The features the reviewer mentions in Fig. 5 are attributed to the TANSO-FTS averaging kernels tending to zero at higher altitudes. Where the comparison is in very good agreement, the TANSO-FTS is approximately equal to its a priori, and the smoothed target profile has been transformed into approximately the a priori.

38.      p. 12 l. 21-22: I find it surprising that measurements within the polar vortex did not impact the results, unless the problematic profiles were discarded through other filters or a priori values were used. Do Holl et al. (2016) apply the same data flags listed in this paper?

Holl et al. (2016) did use the same data quality flags, those for for ACE-FTS are referred to in their manuscript. The primary reason the effect is negligible here is that the temporal duration of the polar vortex is small in our data set.

39.      p. 12 l. 28-29: Following on comments made on the manuscript's introduction, an explanation of why zonal biases may exist should be included. l. 29-31/Figure 5: Reiterating the last general comment, units of ppbv in the left-most panel would remove some of the extra text on the axes associated with the decimals and might be more intuitive for CH4.

Statements explaining this were added to the introduction, where Holl et al. (2016) is first discussed.

40.      p. 12 l. 31-32: Do the sizes of the bins alter the comparisons? Are these zonal ranges narrow enough in the Northern Hemisphere? i.e. Are the profile differences at 50-60N comparable to 30-40N?

The $x$-axes of the left-most panels of Fig. 5 have been changed from ppmv to ppbv.

41.      p. 12 l. 37: Please give a brief explanation of what this statistic tells us (or justification for using it) as opposed to the general correlation coefficient and/or include a citation.

Yes, the bins are narrow enough. Shrinking the zonal bin width results in a paucity of data over the tropics, and we tested different bin sizes and found no significant biases in any zone.

42.      p. 13 l. 18-22: The pressure level at which CH4 decreases is the tropopause height. Unless I am misunderstanding this paragraph, the implication is that the tropopause heights of the instruments are different, which would very likely account for at least some of the profile differences observed. How do the calculated the tropopause heights compare among the various instruments? If they differ, it would indicate that some of the assumptions underlying the pressure interpolation (outlined in the paragraph on p. 11

The Pearson correlation coefficient is the most common form of correlation coefficient and most appropriate for this type of study. The proper name distinguishes it from rank, distance, weighted, adjusted, and other forms of correlation coefficient. What the reviewer refers to as a "general correlation coefficient" is most likely the Pearson correlation coefficient.

43.      p. 13 l. 25-29) might need to be reconsidered. Measurements for which the a priori values have a significant influence could be especially susceptible to tropopause height biases.

We believe the observed phenomenon does not represent the physics of the atmosphere. Since the TANSO-FTS averaging kernels fall off at around the tropopause height, its retrievals have low sensitivity at that altitude. The TANSO-FTS a priori may not accurately reflect the tropopause height, and therefore, neither will the retrievals. Conversely, the other instruments are capable of measuring the height of the tropopause, more accurately defined by the temperature minimum. We avoided discussing the physics of the atmosphere in this section, because we believe the artifact is due more to the fall off of the TANSO-FTS sensitivity and that the vertical resolution of all instruments is not good enough to perfectly reflect the shape of the $CH_4$ VMR vertical profile at the tropopause. The tropopause heights of the instruments tend to agree within uncertainty, but when, for example, ACE-FTS only has observations every 3 or 5 km, that uncertainty is necessarily large. We are not willing to say that the tropopause heights are different because of this feature. We agree that differences in tropopause height pose significant challenges, but the most relevant argument for using pressure instead of tangent altitude is that the tangent altitudes for TANSO-FTS are not know and computing them introduces additional errors. We were careful to say "this feature indicates that the altitude at which this VMR decrease differs" and not that the tropopause heights differ.

44.  p. 13 l. 24/28: By "below 90 hPa" do you mean less that 90 hPa or at lower altitudes? Similarly, does "Above 100 hPa" mean greater than 100 hPa, or at higher altitudes? Looking at the figure, the reader can deduce the appropriate answer but would benefit from less confusing wording.

Referring to pressure is correct in this context since we are discussing a figure with pressure on the axis. We have reworded the statements here to refer to levels.

45.  p. 13 l. 24-27: This result seems to me as one of the most important in the manuscript and deserves elaboration. Does the variability have any notable features? Does it depend on sensitivity (s) or a priori influence? Did you find covariance with latitude (i.e. within the 30 degree bins), tropopause height, or season?

This is explored in Sect. 8. We specifically measure the relationship on mean VMR difference with latitude.

46.  p. 13 l. 33-34/Figure 6: Do zonal differences exist in the unsmoothed data? How do the unsmoothed data fit into the goal mentioned in the introduction for assessing the applicability of Holl et al. (2016) to lower latitudes?

The unsmoothed data didn't exhibit significant zonal differences, which is why we chose to show fewer figures. An explicit mention of the lack of zonal differences has been added to the text.

47.  p. 13 l. 35-36: This sentence is confusing due to the the vague phrasing and verbosity (e.g. "actual differences one would expect"). Please reword.

The first half of this sentence has been re-written to read: "Fig. 6 shows the mean differences between the TANSO-FTS data product and those of other instruments."

48.  p. 13 l. 37: More consistent across instruments? Or across altitudes for each instrument?

The sentence is re-worded to explicitly name differences as those profiles shown in Fig. 6.

49.    p. 14 l. 9-11: Why would the differences between ACE-FTS and ESA MIPAS be smaller? Does this shed light on differences between each of these instruments and TANSO?

The difference between ACE-FTS and ESA MIPAS is smaller, and expectedly so, because both instruments' data products are more accurate throughout the altitude range shown due their increased sensitivity and decade-long retrieval development.

50.    Section 7.1: Does any of the methodology apply to Section 6 (e.g. criteria for minimizing the dominance of the prior) and vice versa?

No, only the selection of coincident measurements and smoothing is common. In Sect. 6, we compare measurements at each pressure level and use whatever pressure levels are available. In Sect. 7, we are integrating the vertical profiles and set a requirement of a minimum overlap of retrieval levels.

51.    p. 14 l. 21-22/Figure 8: The way this is described, it seems contradictory with the caption on Fig. 8, "The vertical range of partial column integration varies for each pair of coincident profiles." If you mean that for each coincident measurement pair you match the vertical range of TANSO and each of the other vertical profiles, but that the vertical ranges across all coincident measurements vary, please describe this more explicitly somewhere in this section. Also, if that interpretation is correct, does the vertical range impact the distributions or correlations of the data? (This same question applies to Figure 9.)

There is some ambiguity here, that we have tried to correct. Each pair compared must have the same integration range. We didn't require every pair to have one range. That the magnitude of the partial columns varies with altitude is discussed on lines 19–24 and on page 15, lines 31–33, and statistics about the variability of altitude ranges are given in Table 3 and discussed on page 16.

52.    p. 14 l. 34: Why a sensitivity threshold of 0.2? This seems a little low. The minimum of three pressure levels also seems low unless they are contiguous (i.e. don't skip filtered out data in the profile). If the data points do not adjoin each other, did you apply criteria on how far apart the levels can be?

This is partly because the sensitivity of TANSO-FTS is this low. The problem isn't just the sample size, however, with a higher threshold, the pairs that remain in the comparison tend to to have short vertical ranges of integration. We are comparing smoothed data which are necessarily interpolated, so the altitudes of overlap are contiguous.

53.    p. 14 l. 5-7: How different are the results when these 23for the outliers in Figures 8 and 9?

There are no outliers, if they are shown, they lie beneath the "good" data. They all have small partial column values because the integration interval is small. This comment is a little confusing because, since we selected a sensitivity threshold, the 23% of the data that does not meet the criteria are excluded by the criteria. A comment about the behaviour of the excluded data has been added to the text.

54.      p. 15 l. 10-20: Given that the values are all on a pressure grid, what is the advantage of integrating in altitude/z versus pressure/P? Do you account for water vapour (i.e. use "dry" P/T)?

Reformulating the problem as an integral of pressure introduces a dependence on the vertical profile of the mass density ($dz = dP/(-\rho g)$). We did not convert to "dry" $P$–$T$ because the coincident measurements are co-located so the humidity should be close to the same, but also, introducing a measurement of water vapour introduces the error in that measurement.

55.      p. 15 l. 27: If partial columns with large gaps in the vertical are included (ref. my comment on p. 14 l.34), an uncertainty related with the interpolation should also be propagated through the calculation.

The data are smoothed and necessarily interpolated. The uncertainty of the interpolated data accounts for the interpolation process.

56.      p. 15 l. 29/Figures 8 and 9: Given the emphasis of the paper on zonal dependencies, please add a colour scale to each plot associated with latitude (bin)?

Unfortunately, colourizing Figs. 8 and 9 with latitudes would not be helpful for our paper. You wouldn't see a latitudinal dependence on the correlation, but, you still see clustering of the data. Since the partial column depends on integration range and the lower limits of the ACE-FTS and MIPAS observations depend on latitude (water vapour, opacity and clouds increasing towards the tropics), then the latitudinal dependence in the partial columns is due to the observation geometry of those two satellites.

57.      p. 15 l. 35-37: What would cause a bias in the intercept? Altitude range? Spectroscopy?

A non-zero, positive $y$-intercept means that the $y$-axis data, TANSO-FTS partial columns, are generally larger, by this amount, than those from other instruments. This is consistent with Fig. 5 which shows that the TANSO-FTS VMRs are generally greater than the other data products.

58.      p. 16 l. 8/Table 3: Why is the minimum altitude for the NDACC measurements so high? Figure 3 indicates that TANSO is at least somewhat sensitive closer to the surface that 3km.

3.3 km is actually very low for a remote sensing instrument. This value not for the NDACC stations, but for the overlap between TANSO-FTS and NDACC obeying our sensitivity and flagging criteria. The pressure maximum in the TANSO-FTS retrievals is always less than that of NDACC, and the sensitivity of TANSO-FTS falls off at the lower altitudes, as shown in Fig. 3e.

59.      p. 16 l. 26: When combining the results, are all data weighed equally, or do you take into account the uncertainties of measurements? Is this the average across all latitudes, or is it a bias that is consistent for all latitudes? Have you also assessed whether altitude related biases exist in the combined data?

The least-squares regression is weighted, since each measurement has a unique uncertainty estimate. Each instrument is treated equally, so that with the largest sample size (MIPAS) has the highest contribution. Notes about the regression

have been added to the text. Altitude-related biases exist due to the averaging kernels of TANSO-FTS. These can be inferred from Fig. 5.

60.          p. 16 l. 28-29: Did you find a bias in the sub-tropics or mid-latitudes?

The bias is latitude dependent; it is the result of weighted least squares regression of all the data, which returned a slope and intercept that are statistically significant. The values discussed are approximations based on the regression results. The magnitude of the bias varies between the tropics and the poles. It is not clear enough in the manuscript that these results are from the least-squares regression and we have attempted to correct that.

61.          p. 16 l. 31-32: It is not clear what type of comparison was done? Regression? ANCOVA?

We compared each parameter with our results, as in Fig. 10, and looked at least squares regression results and correlation statistics. This has been specified in the test.

62.          p. 17 l. 13: The mismatch in vertical extent you point out seems to indicate that these other satellites are not appropriate/useful for validation. If this is not your argument, please rephrase this sentence to make the argument more clear.

The vertical mismatch is unfortunate, and MIPAS and ACE-FTS are not the perfect data sets to compare to, but finding a comparable, satellite-based instrument with the same data product and an overlapping operation timeline is not so easy. This problem is also a result of our TANSO-FTS analysis and comparison. The sixteen ground-based observatories do have fully overlapping altitude ranges. We agree that this sentence and the preceding one are poorly written and do not convey our intent, so they have been rephrased.

63.          p. 17 l. 15: Have you tried smoothing the TANSO profiles to NDACC to see if the agreement is robust?

Because NDACC has a finer vertical resolution than TANSO-FTS TIR, and is sensitive to more vertical structure in the profiles, this operation may not result in "smoothing." Furthermore, since the TANSO-FTS sensitivity is low and its retrieved profiles are close to the a priori over most of its altitude range, the result would be mostly mixing the TANSO-FTS a priori and the NDACC a priori together. Holl et al., (2016) measured the vertical resolution for each pair of measurements to determine which was the lower resolution instrument and in a small number of cases smoothed the TANSO-FTS profiles with the NDACC averaging kernels and a priori. We felt it was more important to maintain a consistent set of steps to create the data sets presented in Figs. 5, 6, 8, 9 and 10.

64.          p. 17 l. 18-19: Given these biases, would you recommend "calibrating" the TANSO retrievals?

This would be case dependant, and because of the small magnitude of the bias, up to the researcher's discretion. Due to the spread of the data shown in Fig. 10 and the uncertainty of each instrument's retrieval, we would not recommend calibrating individual profiles, but would recommend that the bias is considered when a large number of profiles are combined.

65.          p. 17 l. 19-21: Please include how this altitude feature varies (or doesn't) with altitude.

We are not certain of what the reviewer is asking us here. The feature is at a specific altitude, it does not vary with altitude, it is not present at other altitudes. We have re-written this sentence to ensure that which profiles are discussed is clear, and that we are discussing Fig. 5 is also clear.

66.          p. 17 l. 25: It is not clear what "taken over altitude and latitude" means; please reword.

This refers to Fig. 10, but is a typo and should read "dependence of the mean differences, taken over altitude, on latitude." This is cumbersome and has been revised.

67.          p. 17 l. 26-27: What improvements are expected in future versions of the retrieval (e.g. priors, spectroscopy)? Based on your results, what would you recommend needs the most/least attention to produce a more accurate data product? Given the limitations of this TANSO product, what applications would it be suited to?

We have asked our co-authors who work on the TANSO retrievals what is next and included some comments in the conclusion about this. The data product is very important because of the altitude range that it covers, which covers the middle and upper troposphere – not covered by the other data products. This differs and compliments ground based observatories in it global spatial coverage. This is mentioned earlier in the conclusion, but maybe not strongly enough. We have added a statement at the beginning of the conclusion pointing out the importance of the data product.

**2.4   Technical suggestions**

Comments referring to the addition or removal of punctuation were included where I thought they might improve readability and are thus suggestions rather than corrections, except in cases where the serial comma in a list needs to be added.

1.          p. 1 l. 3: Change "CO2 and CH4" to "carbon dioxide (CO2) and methane (CH4)"

We added definitions for $CH_4$ and $CO_2$.

2.          p. 1 l. 15: Change "examine" to "examining"

The subject in this list is "we," so examine is correct here. We have added an "and" to better join clauses.

3.          p. 2 l. 3-6: Add VMR (parts per notation) in parentheses after listed percentages.

Added $\pm 40\,\mathrm{ppbv}$ in parentheses.

4.          p. 2 l. 5: Change "investigated" to "investigate"

Made the correction, thank you.

5.          p. 2 l. 7: Unless "over the equator" is specifically what is meant, change to "in the tropics"

Made the change.

6.         p. 2 l. 29: Remove the comma after "local"

Made the change.

7.         p. 2 l. 33: Change "made in coincidence" to "coincident"

Made the change.

8.         p. 3 l. 30: Add a comma after "surface temperature"

Made the change.

9.         p. 3 l. 32: Add a comma after "(Maksyutov et al., 2008; Saeki et al., 2013)"

Made the change.

10.        p. 4 l. 9: Remove dash after "ACE"

This is the ACE-MAESTRO instrument.

11.        p. 4 l. 19: Add comma after "Boone et al. (2005)"

Made the change.

12.        p. 4 l. 24: Remove comma after (Picone et al., 2002)"

Made the change.

13.        p. 4 l. 25: The use of "and" does not seem correct. Could be replaced with "assimilated into" or "from" depending on the relationship between the met data and the model.

Changed "and" to "'with."

14.        p. 4 l. 30: Add comma after "profiles"

Made the change.

15.        p. 4 l. 35: Add comma after "Odin"

Left as is.

16.        p. 5 l. 7: Awkward placement of "(inclination of 98)"

This has been removed.

17.        p. 5 l. 9: Add comma after "cloud parameters"

Made the change.

18.        p. 5 l. 11: Add comma after "2004"

Made the change.

19.	p. 5 l. 12: Reword end of this sentence, e.g. change ", but" to "with"

Changed to "but with."

20.	p. 5 l. 24: Add comma after "limb scan"

Made the change.

21.	p. 5 l. 26: Add comma after "temperature"

Made the change.

22.	p. 6 l. 20: Change the commas around "below 25km" to parentheses.

Made the change.

23.	p. 7 l. 7: Rephrase "dynamical nature" to a more precise term.

Changed it to "characteristic atmospheric dynamics over Antarctica."

24.	p. 7 l. 15-16: The use of and phrasing after the semi-colon is awkward and makes the sentence unclear.

This is the correct use of a semi-colon, separating two list items which contain commas within them.

25.	p. 8 l. 11-12: The parenthetical is awkwardly worded; please revise for clarity.

This sentence has been revised for clarity.

26.	l.12: Change "differences is also" to "differences are also"

Correction made.

27.	p. 8 l. 13: Please reword "When examining dates with several measurements" to make more clear.

Revised to "several measurements made on the same day."

28.	p. 8 l. 16-17: The grammatical structure of this sentence is difficult to follow. Please reword.

Revised this sentence.

29.	p. 9 l. 32: Add comma after "differ"

Made the change.

30.	p. 10 l. 10: Change comma after "kernel" to semi-colon.

Replaced with a period.

31.	p. 10 l. 13: Add comma after "role"

Made the change.

32.        p. 10 l. 20: Add comma after "altitudes"

Left as is.

33.        p. 10 l. 22: Change "70km. This is shown in Fig. 3e." to "70km (Fig. 3e)."

Moved this clause earlier in the text.

34.        p. 10 l. 22: Add comma after "development"

Made the change.

35.        p. 10 l. 27-28: Please reword. The structure of this sentence is difficult to follow, e.g. the verb ("are shown") appears twice and each has its own modifier.

This sentence had remnants of earlier changes, typos fixed, and further revisions made, thank you.

36.        p. 11 l. 22-23: Remove parentheses, and add a period or semi-colon after "retrieval"

Made the change, using a period.

37.        p. 11 l. 24: Please state which retrieval you're referring to (seems like the higher-resolution profile, but not self-evident).

Revised this clause.

38.        p. 11 l. 27: Remove comma after "equilibrium"

Left as is.

39.        p. 12 l. 21-22: Change "looking for" to "filtering" and remove ", and then filtering these events," to make the sentence more clear.

Made the change.

40.        p. 13 l. 20: I think a verb is missing after "VMR decrease" (e.g. "occurs").

Yes, there was a missing word.

41.        p. 13 l. 33: Add a comma after "zonally"

Made the change.

42.        p. 16 l. 29: Change "or" to "and" ("0.014 ppmv or 0.020 ppmv"), and "Pole" should be plural.

Made the change.

43.        p. 17 l. 10: Remove commas after "sensitivity" and "product"

Made the change.

44.         p. 17 l. 12: Remove comma after "altitudes"

Left as is.

45.         p. 17 l. 12-13: Please reword "and that there is a limitation on the useful upper altitude of its data product of below 15 or 20km" to follow the clarity and structure of the beginning of the sentence.

This has been revised.

46.         p. 17 l. 14: Add "upper" before "troposphere" (Without the addition, this sentence is misleading.)

Made the change.

47.         p. 17 l. 20: Phrase starting with "and in a consistent manner" needs rewording for clarity.

This has been revised.

48.         Figures 2 and 10: The legend has two icons for every instrument, which adds extra visual clutter.

Left as is.

49.         Figure 3: Several line colours do not appear in the legends of a-d. Instead of using a legend, you might consider labelling each line with the pressure using the same colour for the text.

There is not enough room to label every profile, so we only labelled 5 kernels for each instrument.

50.         Figure 7: A heat map or similarly sequential colour scheme could be more helpful for this type of plot.

The bin widths are important here, smoothing the data would not really be helpful.

51.         Figure 8: The "R" is missing on the R2 line of each sub-figure, and it looks like some other letters and numbers might also be missing.

These, and other characters in other figures were stripped from the pdf files when the AMT header was added. We will have to work with editors to ensure these characters remain in a revised manuscript.

52.         Figure 10: The degree symbol is missing between parentheses on the x-axis label. Also, please add additional tick marks on the x-axis. You might consider including light gray grid lines behind the data.

Left background as is, will talk to editors about missing symbols.

**3 List of changes**

Page and line numbers refer to the AMT Discussions paper.

- p. 1, line 13: "vertical profiles are smoothed using the TANSO-FTS averaging kernels." changed to "and smoothing is applied to ACE-FTS, MIPAS, and NDACC vertical profiles. Smoothing in needed to account for differences between the vertical resolution of each instrument and differences in the dependence on a priori profiles. The smoothing operators use the TANSO-FTS a priori and averaging kernels in all cases."

- p. 1, line 15: changed "examine..." to "and examine..."

- p. 2 line 3: changed "...4 % ..." to "...4 % ($\pm \sim 40\,\mathrm{ppbv}$)..."

- p. 2 line 5: changed "...investigated ..." to "...investigate ..."

- p. 2 line 7: changed "...equator ..." to "...tropics ..."

- p. 2 line 12: changed "...$CO_2$..." to "...carbon dioxide ($CO_2$)..."

- p. 2 line 15: added the sentences: "In this work we compare TANSO-FTS measurements with those made by similar instruments in order to validate its quality. Any biases in the data product need to be well understood for it to be used by other researchers, and their discovery may lead to improvements of future versions."

- p. 2 line 16: removed "...between 0.37 and 1.6 μm, each around 0.02 μm wide and chosen to avoid $H_2O$ and $O_2$ absorption. TANSO-CAI..."

- p. 2 line 12: changed "...$CH_4$..." to "...carbon dioxide ($CH_4$)..."

- p. 2 line 24: changed "...(PEARL) in Eureka, Canada" to "(PEARL) at 80° N in Eureka, Canada (Batchelor et al., 2009)."

- p. 2 line 26: added reference (Kurylo and Zander, 2000) after "...(NDACC)."

- p. 2 line 28: removed paragraph break

- p. 2 line 29: removed comma after "...local"

- p. 2 line 33: changed "...made in coincidence ..." to "...coincident ..."

- p. 2 line 30: removed sentence beginning "Any biases..."

- p. 3 line 6: inserted the paragraph: "The question we are asking in this validation study is not *what is the magnitude of the difference between TANSO-FTS retrieved* $CH_4$ *vertical profiles*, but: *given the vertical resolution, information content, and a priori dependence of TANSO-FTS, would* $CH_4$ *vertical profile retrievals derived from another co-located*

*instrument's measurements agree with those for TANSO-FTS*? To answer this question a smoothing operator is applied to the vertical profiles of the instruments with finer vertical resolution (and therefore finer structure in the vertical profiles). This smoothing operator, described by Rodgers and Connor (2003), and presented in Sect. 6.1, uses the a priori profiles and averaging kernels from TANSO-FTS. However, results without smoothing are also presented here, as they will be of interest to data-users. provide an indication of the magnitude of these effects."

– p. 3 line 8: inserted "in order to account for the structure intrinsic to a finer-resolution instrument" after the bracketed clause

– p. 3 line 22: replaced "12900–13200 $cm^{-1}$, 5800–6400 $cm^{-1}$, 4800–5200 $cm^{-1}$, and 700–1800 $cm^{-1}$. The fourth band is in the TIR and is used …" with "the TIR band is between 700–1800 $cm^{-1}$ and is used …"

– p. 3 line 27: changed "coverage" to "spatial coverage"

– p. 3 line 28: changed "…methodology…" to "…nonlinear maximum a posteriori method used for…"

– p. 3 line 30: added comma after "…surface temperature…"

– p. 3 line 32: added comma after reference to Saeki et al., (2003)

– p. 3 line 33: added reference (Saitoh et al., 2009)

– p. 4 line 8: removed "…a circular…"

– p. 4 line 8: removed "…with an inclination of 74° …"

– p. 4 line 33: changed "…ACE-FTS v2.2 data…" to "…ACE-FTS v2.2 $CH_4$ data…"

– p 4. line 10: removed "SCISAT also carries the ACE-Measurement of Aerosol Extinction in the Stratosphere and Troposphere Retrieved by Occultation (MAESTRO) instrument, a dual spectrophotometer with a wavelength range of 285–1030 nm and a spectral resolution of 1–2 nm"

– p 4. line 25: changed "…and their Global…" to "…with their Global…"

– p. 4 line 30: inserted comma after "…$CH_4$ profiles…"

– p. 4 line 32: added "(HALOE)" after the acronym's definition.

– p. 5 line 5: added the sentence: "Waymark et al., (2013) found a slight reduction in $CH_4$ VMR in the v3.0 data near 23 km, and a larger reduction of around 10% between 35–40 km."

– p. 5 line 7: removed "(inclination of 98°)"

– p. 5 line 9: inserted comma after "…cloud parameters…"

– p. 5 line 11: inserted comma after "...2004..."

– p. 5 line 12: changed "but..." to "but with..."

– p. 6 line 20: changed commas surrounding "below 25 km" to parentheses

– p. 6 line 30: changed "references, some information for each instrument" to "spectral range and resolution, and references."

– p. 6 line 7: changed "...dynamical nature of the Antarctic atmosphere" to "...characteristic atmospheric dynamics over Antarctica"

– p. 7 lines 12-16: The first paragraph of Section 3 was re-written to read: "To provide context for the VMR differences found when comparing each instrument to TANSO-FTS, shown in Sect. **??**, we have examined the variability of retrievals made for each instrument. We are interested in determining whether the mean differences found when comparing TANSO-FTS to another instrument are comparable to the differences found when comparing pairs of retrievals for a single instrument. Each pair of observations compared in this study are made at different times and locations and subject to instrument noise and analysis errors. Examining the variability within each data set provides an indication of the magnitude of these effects. Because the observation geometries and rates of spectral acquisition are different for each instrument, our internal comparisons differ for each instrument. For example, TANSO-FTS and MIPAS have a much higher data density than ACE-FTS, which only makes two sets of observations per orbit."

– p. 7 line 18: replaced the sentence beginning with "To examine the..." with "TANSO-FTS vertical profiles tend to be similar to their a priori and, therefore, to each other. To provide context for our validation results, we computed the magnitude of the mean differences between the TANSO-FTS retrievals and their a priori. This is indicative of the instrument sensitivity discussed in Sect. **??** and shows by how much the retrievals deviate from the a priori."

– p. 7 line 20: replaced "...3000 TANSO-FTS..." with ...3000 randomly selected TANSO-FTS..."

– p. 7 line 25: replaced "...ACE-FTS sunset/sunrise measurement in a year..." with "...retrieved profile from an ACE-FTS sunset/sunrise (occultation direction)..."

– p. 7 line 26: replace "...the sunset/sunrise measurement acquired on..." with "...that from..."

– p. 7 line 27: added "(which are in different hemispheres)" to "...sunrise occultations,..."

– p. 7 line 27: changed "...acquisition was not made during an orbit" to "...acquisition was not recorded during a subsequent orbit."

– p. 7 line 30: added "VMR" before "difference"

– p. 7 line 30: removed "between pairs of VMRs"

- p. 7 line 35: changed "are" to: "were made"

- p. 7 line 35: removed "set of"

- p. 7 line 26: added the sentence "This provides an indication of the impact of different retrieval algorithms on retrieved profiles."

- p. 7 line 38: changed "has" to "have"

- p. 8 line 9: changed "we considered the ..." to "we compared pairs of observations made at an NDACC site on the same day. We considered only..."

- p. 8 line 10: removed "...and found a subset of NDACC measurements for each site that were made on the same day."

- p. 8 line 10: removed "then"

- p. 8 line 11: moved "...for each pair of measurements ..." to the beginning of the sentence

- p. 8 line 11: replaced "...made on the same day..." with "...on the standard NDACC retrieval grid..."

- p. 8 line 11: replaced "...multiple profiles" with "multiple coincidences in a day"

- p. 8 line 12: removed "the" and "all" from "...the differences are all found..."

- p. 8 line 13: changed "...dates with several measurements" to "...several measurements from the same day"

- p. 8 line 16: removed "Of the four data sets"

- p. 8 line 17: changed "and MIPAS..." to "that MIPAS..."

- p. 8 line 17: changed "while NDACC..." to "and that NDACC..."

- p. 8 line 17: replaced the sentence beginning "The magnitude of..." with "The magnitude of the internal variability of the data sets is between $\pm 2\,\mathrm{ppbv}$ (e.g., for NDACC and ACE-FTS in the upper troposphere) and $\pm 3\,\mathrm{ppbv}$, or around $2\,\%$ (e.g., for TANSO-FTS and the lower limits of ACE-FTS)."

- p. 8 line 21: replaced "In the case of ACE-FTS, which only records two occultations per orbit, and NDACC stations, which are stationary, the objective of the coincidence criteria was to maximize the number of measurements used. Conversely, in the case of MIPAS, which makes frequent observations, the objective was to reduce the number of potential coincident measurements." with "ACE-FTS has an inclination of $74°$ and operates in solar occultation mode, recording only two occultations per orbit, predominantly at high latitudes; the NDACC sites are stationary; MIPAS makes frequent observations at all latitudes; and the spatial distribution of TANSO-FTS observations is enhanced by its cross-track observation mode. In the case of ACE-FTS and NDACC stations, the objective of the coincidence criteria was to maximize the number of measurements used. Conversely, in the case of MIPAS, the objective was to reduce the number of potential coincident measurements."

– p. 8 line 26: inserted reference (Vincenty 1975)

– p. 8 line 28: the following paragraph was added: "The criteria used in this study are comparable to previous $CH_4$ validation studies. For example, de Mazière et al. (2008) used criteria of 24 hours and 1000 km when comparing ACE-FTS $CH_4$ to ground sites, and 6 hours and 300 km when comparing ACE-FTS to MIPAS. Payan et al. (2009) used criteria of 3 hours and 300 km when comparing MIPAS $CH_4$ to ground- and satellite-based spectrometers. Laeng et al. (2015) used criteria of 9 hours and 800 km when comparing MIPAS $CH_4$ to ACE-FTS, and 24 hours and 1000 km when comparing MIPAS to HALOE.

– p. 9 line 9: added reference to Holl et al. (2016) to the end of the sentence ending "...of the means."

– p. 9 line 19: changed "$z$ score" to "standard score"

– p. 9 line 32: inserted comma after "...data set differ..."

– p. 9 line 33: changed "...with widths indicative of the..." to "...whose full-width at half maximum (FWHM) can be used to define the..."

– p. 10 line 6: changed "Each panel shows the mean from 30 retrievals, with each averaging kernel interpolated to a common pressure grid for that instrument" to "Each panel shows the mean from 30 retrievals. Vertical profiles of pressure associated with each retrieval's averaging kernel matrix are, in general, unique, so a common pressure grid was selected for each instrument and averaging kernels were interpolated prior to averaging."

– p. 10 line 10: replaced semi-colon with a period

– p. 10 line 13: inserted comma after "...role..."

– p. 10 line 22: changed "...ACE-FTS and MIPAS is close ..." to "...ACE-FTS and MIPAS, shown in Fig. **??**e, is close..."

– p. 10 line 23: removed the sentence "This is shown in Fig. 3e."

– p. 10 line 22: inserted comma after "...development..."

– p. 10 line 27: replaced the sentence beginning "The trace of the ..." with "The trace of the averaging kernel matrix gives the DOFs. For example, DOFs for retrievals made by TANSO-FTS, IMK-IAA MIPAS, ESA MIPAS, and NDACC from observations over the Arctic, above $60°$ N, are shown in Fig. **??**."

– p. 10 line 32: inserted the sentence "The trends visible are seasonal and are related to opacity and water vapour content."

– p. 11 line 22: removed parentheses around clause beginning "$\mathbf{x}_a$ and $\mathbf{A}$ ..."

– p. 11 line 24: changed "or the retrieval ..." to "or when the retrieval ..."

- p. 11 line 28: added reference to Sepúulveda et al. (2014).

- p. 11 line 29: changed "...common pressure grid" to "...common pressure grid, as opposed to an altitude grid."

- p. 11 line 30: inserted the sentence "Extrapolation is needed to ensure that the length of $\hat{x}$ matches the dimensions of $\mathbf{A}$ in Eq. 1.

- p. 12 line 21: changed "...looking for..." to "...identifying and removing..."

- p. 12 line 22: removed "and then filtering these events"

- p. 12 line 23: changed "affect" to "effect"

- p. 13 line 20: changed "...VMR decrease differs..." to "...VMR decrease occurs differs..."

- p. 13 line 24: changed "...below 90 hPa" to "...below the

- p. 13 line 28: changed "Above 100 hPa..." to "Above the 100 hPa level..."

- p. 13 line 24: inserted the sentence "No zonal biases were observed in the unsmoothed data."

- p. 13 line 33: inserted comma after "...zonally..."

- p. 13 line 35: changed "This study reveals the actual differences one would expect when using the TANSO-FTS data product" to "Fig. 6 shows the mean differences between the TANSO-FTS data product and those of other instruments"

- p. 13 line 37: changed "the differences..." to "the difference profiles in Fig. 6..."

- p. 14 line 21: changed "For consistency, partial columns..." top "For consistency, each pair of partial columns..."

- p. 14 line 23: changed "...partial columns" to "...for each coincident pair of profiles"

- p. 14 line 37: inserted the sentence "These excluded data do not exhibit a broader distribution, but their computed partial columns are all very small due to the integration range."

- p. 15 line 24: changed "0.9986, 0.9968, 0.9965, and 0.9929 for ACE-FTS, ESA MIPAS, IMK-IAA, MIPAS, and NDACC" to "0.9986, 0.9965, 0.9968, and 0.9958 for ACE-FTS, IMK-IAA MIPAS, ESA MIPAS and NDACC"

- p. 16 line 26: the sentence beginning "A bias is seen..." has been replaced with "Weighted least squares regression of the combined data sets for each hemisphere reveals a bias at all latitudes of $13.30 \pm 0.06$ ppbv."

- p. 16 lines 26, 28, 29: changed units from ppmv to ppbv

- p. 16 line 28: changed "...combined data set..." to "...combined data sets in each hemisphere..."

- p. 16 line 29: changed "...or..." to "...and..."

- p. 16 line 30: added the sentence "The biases are latitude-dependent and vary between the tropics and the poles."

- p. 16 line 32: inserted the sentence "Each parameter was compared to the latitudes and the mean differences in Fig. 10, and the regression and covariance statistics from least squares fitting were computed."

- p. 16 line 38: added the following paragraphs to the discussion:

The primary driver of the mean differences found when comparing TANSO-FTS to other FTS instruments, with and without smoothing, is the instrument design and observation geometry. TANSO-FTS is a much more compact and, therefore, coarser spectral resolution FTS than those used in the comparison. The coarser spectral resolution makes it harder to distinguish closely spaced absorption lines, leading to poorer vertical sensitivity and higher uncertainty in the measurements. While the TIR spectral range of TANSO-FTS is comparable to that of MIPAS, the mid-infrared ranges of NDACC and ACE-FTS include a very strong methane absorption band near $3000 \text{ cm}^{-1}$ with little interference from $CO_2$, increasing their sensitivity and ability to accurately constrain $CH_4$ retrievals. Furthermore, MIPAS and ACE-FTS observe the limb of the atmosphere, providing them with more measurements per retrieved profile, improved vertical resolution, and much higher sensitivity. While NDACC instruments also only have a single spectrum per retrieved profile, they observe the sun directly (as does ACE-FTS), resulting in a very strong signal. All these factors contribute to TANSO-FTS performing retrievals on a lower spectral resolution measurement of a weaker signal compared to MIPAS, ACE-FTS and the NDACC sites. This results in the sensitivity and DOFs shown in Figs. 3 and 4.

In Sect. 3, we examined the variability within each data set. This gives an idea of some of the sources of error in our comparison. The coincidence criteria used allow for the comparison of retrieved $CH_4$ vertical profiles from different air masses. Our investigation of the NDACC data provides an estimate of the dependence of the $CH_4$ abundance on time, since we compared profiles retrieved from the same location using the same retrieval algorithms, but at different times of day. Our result shows that temporal spacing may contribute around 5 ppbv. Our investigation of the ACE-FTS variability fixed the instrument and retrieval algorithm, but compared observations of different air masses, and we found a similar result of only several ppbv. The largest variability was exhibited when we investigated the MIPAS data set. This comparison was of the same observations analyzed by different retrieval algorithms (IMK-IAA and ESA), and resulted in much larger mean differences on the order of 100 ppbv.

Differences in retrieval algorithms between TANSO-FTS and the validation instruments may also account for the differences found in Figs. 5 and 6. Small differences in spectroscopic parameters exist, for example, each instrument's retrieval algorithms use different editions of the HITRAN line list. Comparisons of these line lists, and their impact on retrievals, can be found in **???**. The most significant parameter for TANSO-FTS is its a priori due to the weight given to the a priori profile by the TANSO-FTS averaging kernels in the retrieval. In Sect. 3 we compared the TANSO-FTS retrieved vertical profiles of $CH_4$ to the corresponding

a priori profile and found that they differ, on average, by up to 30 ppbv. This provides a rough minimum of the accuracy of the a priori profiles required for the the retrievals.

– p. 17 line 10: Inserted the sentences "The TANSO-FTS TIR $CH_4$ vertical profile data product is an important and novel data set. Its vertical range extends lower into the troposphere than other satellite data products, and its spatial coverage is global with a high density of measurements."

– p. 17 line 10: removed comma after "...the sensitivity"

– p. 17 line 10: removed "...$CH_4$ TIR vertical profile..."

– p. 17 line 12: removed "useful"

– p. 17 line 12: changed 'below" to "around"

– p. 17 line 13: replaced the sentence beginning "However,..." with "Unfortunately, the lower altitude boundaries of the other satellite-based data products, between 7–15 km, reduces the vertical range over which we can make comparisons."

– p. 17 line 14: changed "in the troposphere" to "in the upper troposphere"

– p. 17 line 19: replaced the sentence beginning "We found that..." with "We found that the shapes of the TANSO-FTS $CH_4$ VMR vertical profiles near 15 km, where the $CH_4$ VMR falls off with increasing altitude, does not match those of the other instruments, and in a consistent manner, resulting in a pronounced feature in the mean difference profiles in Fig. 5, just below the 100 hPa level."

– p. 17 line 25: changed "...dependence of the mean differences, taken over altitude and latitude" to "...dependence of the vertically-averaged differences on latitude."

– p. 17 line 26: Inserted paragraph break before the sentence starting "We look..."

– p. 17 line 27: Inserted the sentences " In a future release, the a priori will not be changed, but remain the outputs of the NIES-TM. Kuze et al. (2016) used theoretical simulations to determine that the Level 1B spectra which were used (V161) to generate the current TIR $CH_4$ data product had considerable uncertainties. New Level 1B spectra are due for release in 2018 and should lead to improved retrievals. Kuze et al. (2016) also proposed some corrections to the TANSO-FTS TIR L1B spectra which may be implemented. The spectral line list used (HITRAN 2008) will be updated. Uncertainties in the surface emissivity over cold surfaces (snow and ice) affect the retrieval at higher altitudes and will be improved in the next release. Improvements are also being made to the way the retrieval handles and simultaneously retrieves interfering species, such as $O_3$."

– p. 19 line 10: changed "formulations" to "formulation"

– p. 19 line 13: updated the Bader et al. (2016) reference to reflect a change from ACPD to ACP

- p. 20 line 19: updated the Errera et al. (2016) reference to reflect a change from AMTD to AMT

- p. 21 line 11: changed "Annal." to "Ann."

- p. 21 line 33: changed "1486" to "1468"

- Table 1: in footnote $^a$ changed ". . . are often. . . " to ". . . may be. . . "

- Table 1: added the footnote "The Altzomoni site came online in late 2012."

[revised manuscript text omitted]

---

## Referee Report (RR1)

Review of "Comparison of the GOSAT TANSO-FTS TIR CH$_4$ volume mixing ratio vertical profiles with those measured by ACE-FTS, ESA MIPAS, IMK-IAA MIPAS, and 16 NDACC stations" by K. S. Olsen et al.

General comments:

The authors have adequately dealt with my points raised following the first review. I recommend publication with minor revisions below.

[1] p1, line13: "Smoothing in needed" ---> "Smoothing is needed"

[2] p3, line16: I would change this sentence as follows.

"However, results without smoothing are also presented here, as they will be of interest to data-users." ---> "In this study, results with and without smoothing are presented (Sect. 7)."

[3] p8, line18: "the IMK-IAA data has" ---> "the IMK-IAA data have"

[4] p10, line30: "the full-width at the half-maximum values" ---> "the FWHM values"

[5] p10, line31: "the full-width at the half-maximum values" ---> "the FWHM values"

[6] p18, line25: "the corresponding a priori profile" ---> "the corresponding a priori profiles"

[7] p20, line7: "NDACC data has" ---> "NDACC data have"

[8] p20, line8: "and was accessed" ---> "and were accessed"

[9] p34: In Figure 5, "N° coincidences" ---> "No. coincidences"

---

## Author Response (AR2)

**Authors' response to: "Comparison of the GOSAT TANSO-FTS TIR $CH_4$ volume mixing ratio vertical profiles with those measured by ACE-FTS, ESA MIPAS, IMK-IAA MIPAS, and 16 NDACC stations"**

Kevin S. Olsen[1], Kimberly Strong[1], Kaley A. Walker[1,2], Chris D. Boone[2], Piera Raspollini[3], Johannes Plieninger[4], Whitney Bader[1,5], Stephanie Conway[1], Michel Grutter[6], James W. Hannigan[7], Frank Hase[4], Nicholas Jones[8], Martine de Mazière[9], Justus Notholt[10], Matthias Schneider[4], Dan Smale[11], Ralf Sussmann[4], and Naoko Saitoh[12]

[1]Department of Physics, University of Toronto, Toronto, Ontario, Canada
[2]Department of Chemistry, University of Waterloo, Waterloo, Ontario, Canada
[3]Istituto di Fisica Applicata "N. Carrara" (IFAC) del Consiglio Nazionale delle Ricerche (CNR), Florence, Italy
[4]Institut für Meteorologie und Klimaforschung, Karlsruhe Institute of Technology, Karlsruhe, Germany
[5]Institute of Astrophysics and Geophysics, University of Liège, Liège, Belgium
[6]Centro de Ciencias de la Atmósfera, Universidad Nacional Autónoma de México, Mexico City, Mexico
[7]Atmospheric Chemistry Division, National Center for Atmospheric Research, Boulder, CO, USA
[8]Centre for Atmospheric Chemistry, University of Wollongong, Wollongong, Australia
[9]Belgisch Instituut voor Ruimte-Aëronomie–Institut d'Aéronomie Spatiale de Belgique (IASB-BIRA), Brussels, Belgium
[10]Institute for Environmental Physics, University of Bremen, Bremen, Germany
[11]National Institute of Water and Atmospheric Research Ltd (NIWA), Lauder, New Zealand
[12]Center for Environmental Remote Sensing, Chiba University, Chiba, Japan

*Correspondence to:* K. S. Olsen (ksolsen@atmosp.physics.utoronto.ca)

We would like to thank both referees and the journal editors for spending their time assessing and reviewing our manuscript. They were very thorough and careful, and we value their input.

**1 Anonymous Referee #1**

The authors have adequately dealt with my points raised following the first review. I recommend publication with minor revisions below.

1. p1, line13: "Smoothing in needed" $\longrightarrow$ "Smoothing is needed"

2. p3, line16: I would change this sentence as follows. "However, results without smoothing are also presented here, as they will be of interest to data-users." $\longrightarrow$ "In this study, results with and without smoothing are presented (Sect. 7)."

3. p8, line18: "the IMK-IAA data has" $\longrightarrow$ "the IMK-IAA data have"

4. p10, line30: "the full-width at the half-maximum values" $\longrightarrow$ "the FWHM values"

5. p10, line31: "the full-width at the half-maximum values" $\longrightarrow$ "the FWHM values"

6. p18, line25: "the corresponding a priori profile" $\longrightarrow$ "the corresponding a priori profiles"

7. p20, line7: "NDACC data has" $\longrightarrow$ "NDACC data have"

8. p20, line8: "and was accessed" $\longrightarrow$ "and were accessed"

9. p34: In Figure 5, "N° coincidences" $\longrightarrow$ "No. coincidences"

Thank you very much for pointing out these remaining errors. All changes have been made as suggested.

**2 List of Changes**

– p1, line13: "Smoothing in needed" changed to "Smoothing is needed"

– p3, line16: "However, results without smoothing are also presented here, as they will be of interest to data-users." changed to "In this study, results with and without smoothing are presented (Sect. 6.3)."

– p8, line18: "the IMK-IAA data has" changed to "the IMK-IAA data have"

– p10, line30: "the full-width at the half-maximum values" changed to "the FWHM values"

– p10, line31: "the full-width at the half-maximum values" changed to "the FWHM values"

– p18, line25: "the corresponding a priori profile" changed to "the corresponding a priori profiles"

– p20, line7: "NDACC data has" changed to "NDACC data have"

– p20, line8: "and was accessed" changed to "and were accessed"

– p34: In Figure 5, "N° coincidences" changed to "No. coincidences"

[revised manuscript text omitted]